# LangNav: Language as a Perceptual Representation for Navigation

## Abstract

We explore the use of language as a perceptual representation for vision-and-language navigation. Our approach uses off-the-shelf vision systems (for image captioning and object detection) to convert an agent's egocentric panoramic view at each time step into natural language descriptions. We then finetune a pretrained language model to select an action, based on the current view and the trajectory history, that would best fulfill the navigation instructions. In contrast to the standard setup which adapts a pretrained language model to work directly with continuous visual features from pretrained vision models, our approach instead uses (discrete) language as the perceptual representation. We explore two use cases of our language-based navigation (LangNav) approach on the R2R vision-and-language navigation benchmark: generating synthetic trajectories from a prompted large language model (GPT-4) with which to finetune a smaller language model; and sim-to-real transfer where we transfer a policy learned on a simulated environment (ALFRED) to a real-world environment (R2R). Our approach is found to improve upon strong baselines that rely on visual features in settings where only a few gold trajectories (10-100) are available, demonstrating the potential of using language as a perceptual representation for learning navigation agents.

## 1 Introduction

Applications of large language models (LMs) to non-linguistic embodied tasks have generally focused on using the implicit world knowledge within LMs to predict sub-tasks and actions for planning (Ahn et al., 2022; Huang et al., 2022b;a; Singh et al., 2022). For instance, recent work has shown that LMs can be prompted to create a list of actions (e.g., `GoToBathroom`, `LocateToothbrush`) given a high-level goal given in natural language (e.g., "brush teeth") (Huang et al., 2022a). These approaches rely on the LM's priors on action sequences and inter-object correlations acquired through large-scale pretraining (Zhou et al., 2023b; Li et al., 2023; Zhao et al., 2023), and it has not been clear whether such text-only models can be adapted to tasks such as vision-and-language navigation which requires an egocentric agent follow instructions to navigate a 3D environment using visual input.

To be clear, there *is* a substantial body of work on using pretrained LMs for vision-and-language navigation tasks (Hong et al., 2021; Qi et al., 2021; Qiao et al., 2022, *inter alia*). The standard approach is to simply use a pretrained LM over the natural language instructions to extract text features that are combined with the agent's perceptual representations, which are given by continuous image features extracted from pretrained vision models (Wang et al., 2019; Hao et al., 2020; Fried et al., 2018). While effective in data-rich regimes, the direct use of vision features makes the approach difficult to apply in cases where only a few labeled trajectories exist (e.g., 10-100 trajectories), as this is typically not enough data to learn a joint vision-language model without overfitting (even with pretrained models). A popular strategy in such data-scarce regimes is to generate synthetic data or transfer knowledge from other domains (e.g., from simulated environments). However, generating realistic perception data is itself a difficult task, and sim-to-real transfer with models that purely rely on visual features is prone to overfitting to the features of simulated environments (Anderson et al., 2021).

This paper proposes an alternative approach for learning vision-and-language navigation agents by exploiting language itself as a perceptual representation space. Our approach uses off-the-shelf vision models to obtain textual descriptions of the agent's egocentric panoramic view. The text descriptions are then fed to an LM which must select the next action given the instruction and (text descriptions of) the previous actions or observations. See fig. 1 for an overview.

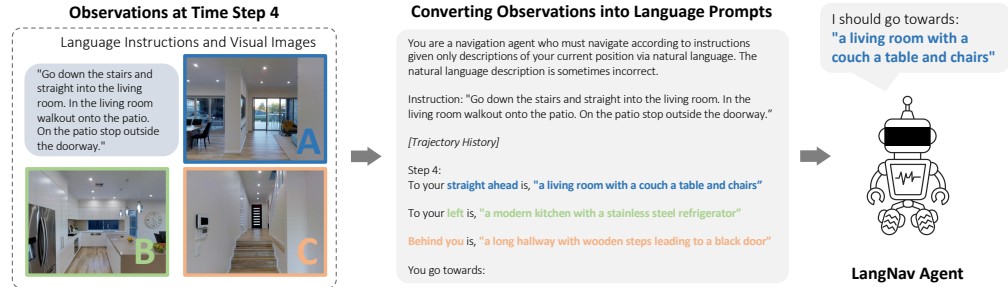

Figure 1: Overview of our proposed LangNav approach. We describe the task instructions and visual observations (from off-the-shelf vision systems) through text. A language model uses pure language descriptions to predict which direction to move towards. Here, views **A**, **B**, and **C** correspond to the front, left, and rear views of the agent.

The use of a discrete language space to represent an agent's perceptual field makes it possible to readily leverage the myriad capabilities of large language models. In our first case study, we show how we can use a small amount of seed training data (10-100 trajectories) to obtain synthetic "trajectories" from a powerful but closed-source LM (GPT-4). We find that training a smaller language model (LLaMA-7B & LLaMA2-7B) on the generated trajectories mixed with the original seed data results in a language-based navigation (LangNav) agent that outperforms a vision-based agent that is finetuned on the same seed data. In our second study, we explore the use of language as a domain-invariant representation to perform sim-to-real transfer, where we transfer an agent trained on a simpler simulated environment (ALFRED; Shridhar et al., 2020) to the real-world R2R (Anderson et al., 2018b) environment. Insofar as language is hypothesized to have co-evolved with the human brain to enable efficient communication (Deacon, 1997), it naturally abstracts away low-level perceptual details, and we indeed find that LangNav exhibits improved sim-to-real transfer compared to the vision-based agent. Our results collectively suggest that using language as a perceptual representation for vision-and-language navigation is feasible and sometimes outperforms traditional approaches that rely on continuous visual features in low data regimes.

## 2 BACKGROUND: ROOM-TO-ROOM VISION-LANGUAGE NAVIGATION

A popular real-world testbed for learning vision-and-language navigation (VLN) agents is the room-to-room dataset (R2R; Anderson et al., 2018b), in which an agent must perceive and navigate a 3D environment based on a language instruction $U$ and an initial state $S_0$. At each time step $t$, the agent uses the current observation $O_t$, the original language instructions $U$, and the trajectory history $H_t$, to predict the panoramic action $a_t$. The current observation is given by a set of panoramic images that describe the agent's egocentric view, i.e., $O_t = \{I_{t,0}, ..., I_{t,V}\}$ where $V$ corresponds to the number of discretized view angles.[1] The panoramic action $a_t$ corresponds to which navigable view in $O_t$ to go towards, i.e., $a_t \in O_t$. After selecting an action, the state transitions from $S_t$ to $S_{t+1}$. The aim is to output the command STOP after reaching the goal $G$ specified by $U$ in state $S_0$.

The standard approach in R2R is to process the panoramic images $\{I_{t,0}, ..., I_{t,V}\}$ with a pretrained visual encoder $E_v$ to extract continuous visual features $F_{t,v} = \{E_v(I_{t,0}), ..., E(I_{t,V})\}$ (Anderson et al., 2018a; Fried et al., 2018; Tan et al., 2019; Hong et al., 2020). The language instruction is typically processed by a pretrained language encoder $E_l$ (e.g., BERT (Devlin et al., 2019)) to extract the language features $F_l = E_l(U)$. These features, along with a hidden state representation of the trajectory history $h_{t-1}$, are fed to a joint vision-language module (e.g., another Transformer) that attends over $\{I_{t,0}, ..., I_{t,V}\}$ to select the action $a_t$.

## 3 LANGUAGE AS A PERCEPTUAL REPRESENTATION FOR NAVIGATION

We begin by describing the perception-to-text models employed for converting visual observations into text (§ 3.1). We then discuss the prompt templates for converting the text into natural language (§ 3.2), followed by a description of the offline imitation learning algorithm for learning (§ 3.3).

### 3.1 VISION-TO-TEXT SYSTEM

We use off-the-shelf vision models to convert visual observations into language descriptions. We use an image captioning model (BLIP; Li et al., 2022a) and an object detection model (Deformable

---

[1]In the popular R2R benchmark this can be as many as 36 (12 headings and 3 elevations). However we follow previous works only consider the navigable views, which is often many fewer than 36.

DETR; Zhu et al., 2020) over each view angle $I_{t,j}$ to obtain the text descriptions,

$$C_{t,j} = \text{IMAGECAPTIONER}(I_{t,j}), \qquad x_{t,j,0}, \ldots, x_{t,j,M} = \text{OBJECTDETECTOR}(I_{t,j}),$$

where $M$ is the number of detected objects.

## 3.2 PROMPT TEMPLATES

Fig. 1 illustrates how the image caption and the detected objects are combined via templates to construct a piece of text on which to condition the language model. Based on the prompt template, the language model will be finetuned on the (language representations of) output actions $\{(a_1), \ldots, (a_T)\}$ via the (conditional) language modeling objective. The prompt consists of the following components. (An example of a full trajectory is shown in appendix E).

**Task description $D$.** We first provide the language-based agent that describes the task:

```
You are a navigation agent who must navigate according to instructions
given only descriptions of your current position [...].
```

**Navigation instruction $U$.** We then give the natural language instruction for the task, which provides guidance to the agent on how to reach the goal. In this paper, the high-level instructions can be from the realistic R2R dataset (our main dataset), synthesized by GPT-4 (which we use for data augmentation), and the ALFRED dataset (from which we perform sim-to-real transfer learning). An example instruction from R2R is:

```
Travel forward past the wall with all the light switches and into the
first room on your right.
```

**Current observation $O_t$.** We use templates to convert the image caption $C_{t,j}$ and objects obtained $x_{t,j,0}, \cdots, x_{t,j,M}$ from $I_{t,j}$ (§ 3.1). For instance, if the agent is facing a heading of 90 degrees and an elevation of 0 degrees and there is a candidate navigable direction $I_{t,j}$ located at a heading of 120 degrees and an elevation of 0 degrees, the text description for this view angle would be:

```
To your 30 degree right is "{C_t,j}".
Details: {x_t,j,0},...,{x_t,j,M}.
```

(These view angles are given by the dataset.) We create such templates for all the navigable view angles $\{I_{t,0}, \ldots, I_{t,V}\}$.

**Action $a_t$.** Selecting an action involves choose a navigable view out of $O_t$ to move towards, i.e., $a_t \in O_t$. For example, suppose $a_t = I_{t,j}$, i.e., the agent decided to go to the $j$-th view angle. Then this is recorded as

```
You go towards: "C_t,j"
```

To actually have the agent generate $a_t$ we simply decode from an LM $p_{\text{LM}}(\cdot \,|\, D, U, H_t, O_t)$ with greedy decoding, where $H_t = \{O_i, a_i\}_{i=0}^{t-1}$ encodes the observation and action trajectory. We found the LM to have no issue generating from the set of navigable directions (i.e., $\{C_{t,0}, \ldots, C_{t,V}\}$) with simple left-to-right decoding, and thus did not need to perform constrained decoding or employ alternative strategies (e.g., run inference multiple times and select the highest scoring action).

**Updating trajectory history $H_t$.** We update the observation and action trajectory history via appending the text representations of $O_t$ and $a_t$ to $H_t$. Specifically $O_t$ and $a_t$ are appended via adding the following template:

```
Step {t}: To your {direction_1} is {caption_1}; To your {direction_2} is
{caption_2}; [...]; You chose: {caption_of_selected_direction}.
```

This history serves to inform the model about its current position within the high-level instruction, enabling it to make more informed decisions when selecting actions.

**Remark.** Due to the nontrivial amount of compute resources required for running our experiments (e.g., generating synthetic data from GPT-4, training a large LM on the generated synthetic trajectories), we did not experiment with the prompt templates too much and just used something that seemed reasonable. Similarly, for our off-the-shelf vision systems we quickly converged on the above two models which seemed to qualitatively produce reasonable results.

### 3.3 IMITATION LEARNING ON DEMONSTRATIONS

The navigation agent is trained via offline imitation learning via finetuning a pretrained language model (LLaMA, Touvron et al. (2023b)) on the above template. Concretely, we create an instruction-following dataset by transforming the expert trajectory from the original dataset into instruction-following demonstrations using the templated approach. Let $\mathcal{D} = \{W^{(i)}\}_{i=1}^N$ be the set of training trajectories, where each $W^{(i)}$ can be represented as a natural language sequence from the above template, $W^{(i)} = (D^{(i)}, U^{(i)}, H_1^{(i)}, O_1^{(i)}, a_1^{(i)}, \ldots, H_{T^{(i)}}^{(i)}, O_{T^{(i)}}^{(i)}, a_{T^{(i)}}^{(i)})$. Here $T^{(i)}$ is the number of actions in the example $W^{(i)}$, which is typically between 5 to 7. Given the above, we optimize the log likelihood of the actions, i.e., the objective for trajectory $W^{(i)}$ is given by,

$$\sum_{t=1}^{T^{(i)}} \log p_{\text{LM}}(a_t^{(i)} \mid D^{(i)}, U^{(i)}, H_t^{(i)}, O_t^{(i)}).$$

While behavior cloning on gold trajectories is simple, it is prone to error propagation. In particular, the history trajectory is obtained by a shortest-path algorithm (which has knowledge of the goal) and thus adheres closely to an optimal policy $\pi^*$. However, during prediction, trajectories can deviate significantly from the optimal policy, leading to a distribution shift that can adversely affect performance. To make sure the trained policy can recover from deviations from the optimal path, we adopt the following strategy to create our imitation learning dataset: (1) at each time step, we sample a random action with probability $\rho = 0.2$; (2) once a random action is selected, we use the shortest-path algorithm to obtain the ground truth next action; (3) we repeat this process until the goal is reached; (4) once the goal is reached, this becomes part of the training demonstration data. While more involved strategies which (for example) sample from the current policy are possible (Ross et al., 2011) (and in fact widely used in the vision-based navigation literature), we found the above to be simple and effective.

## 4 LANGNAV: EMPIRICAL STUDY

Our primary experiments target the low-data setting, motivated by the observation that obtaining annotated data for embodied tasks such as vision-language navigation is often very costly (often more so than text-only or vision-only tasks). In particular, we are interested in learning the most performant system based on a small number (10 or 100) of real-world trajectories. We sample our real-world trajectories from Room-to-Room (R2R) dataset (Anderson et al., 2018b), a realistic vision-and-language navigation dataset consisting of 21,567 navigation instructions in the Matterport3D Anderson et al. (2018b) environment. The dataset includes 90 scenes, with 61 scenes in the train and validation "seen" sets, and 11 scenes in the validation "unseen" set. Our 10-shot dataset is randomly sampled the train set within 1 scene, while our 100-shot dataset spans 2 scenes.

**Evaluation.** To contextualize our approach against prior work, we evaluate LangNav on both "seen" and "unseen" sets from R2R. The "seen" set contains scenes identical to the training set (but the instructions and trajectories differ). However, this distinction is less important for our low-data regime, since we only make use of 1 scene (for the 10-shot case) or 2 scenes (for the 100-shot case). I.e., the majority of scenes in the "seen" validation subset has actually been unexposed to the agent.

For evaluation, we use the standard R2R task performance metrics (Anderson et al., 2018a). *Navigation Error* (NE), the average distance between the agent's final position and the goal in meters (lower is better); *Success Rate* (SR), the ratio of trajectories in which the agent stopped within 3 meters of the goal (higher is better); *Oracle Success Rate* (OSR), the ratio of trajectories in which the agent stopped within 3 meters to the goal with a view of the goal (higher is better); and *Success* weighted by the normalized inverse of the *Path Length* (SPL) (higher is better).

### 4.1 CASE STUDY 1: LANGUAGE ENABLES EFFICIENT SYNTHETIC DATA GENERATION

In NLP, obtaining synthetic data from an appropriately-prompted large language model with which to learn a smaller model has been shown to be an effective approach in data-scarce settings (Wang et al., 2021; Lang et al., 2022; Taori et al., 2023; Dai et al., 2023; Gunasekar et al., 2023, *inter alia*).[2] However this approach is difficult to extend to non-linguistic perceptual tasks such as vision-language navigation since generating realistic perception data is itself a difficult task. In this section we show

---

[2]However see Gudibande et al. (2023) for a critical discussion of this approach.

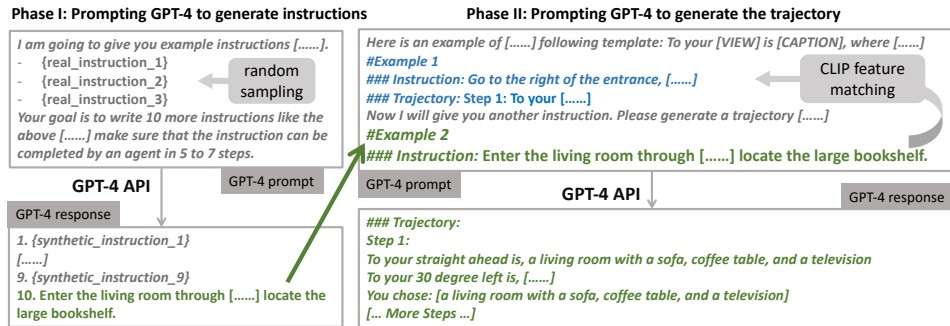

Figure 2: Pipeline for generating trajectories from a prompted GPT-4. In Phase 1, we prompt GPT-4 with 3 randomly sampled navigation instructions $U$ to generate 10 more synthetic navigation instructions. Then in Phase 2, for each generated navigation instruction, we prompt GPT-4 to generate the trajectory that fulfills the generated instruction. See appendix F for details.

that working in pure language space makes it possible to easily generate high quality synthetic data from a large language model based on a few seed trajectories. We further show LangNav, which is trained on a mixture of synthetic and real trajectories, outperform vision-based agents, when the latter is trained on the 10-100 real trajectories.

### 4.1.1 SYNTHETIC TRAJECTORY GENERATION

We generate the synthetic trajectories by using only the 10-shot real-world trajectories from a single scene (see § 4). In R2R each real trajectory has 3 navigation instructions which are narrated by 3 different annotators. Thus we have 30 navigation instructions $\{U^{(i)}\}_{i=1}^{30}$ in total. Our data generation pipeline can be divided into two phases. In phase 1, we randomly choose 3 real instructions as prompt examples and ask GPT-4 to create 10 more instructions similar to the examples, as is shown in fig. 2. We collect 10,000 generated navigation instructions in this phase. In phase 2, for each generated instruction, we prompt GPT-4 to generate a trajectory to fulfill the instruction, conditioned on a real demonstration instruction and trajectory. The real trajectory is obtained by selecting the trajectory whose instruction is closest to the synthetic instruction based on the CLIP (Radford et al., 2021) text features. See fig. 2 for an overview and appendix F for the GPT-4 prompts.

We present an illustrative example in fig. 3 to demonstrate the characteristics of the generated trajectories. Following the pipeline depicted in fig. 2, we first generate an instruction, such as "Enter the hallway [...]" and then prompt GPT-4 to generate a trajectory that fulfills the given instruction. We find three key aspects that indicate the quality of the generated trajectories: **1. Strong Prior:** The generated scenarios exhibit a strong adherence to real-world room-object and object-object correlations, as evident from descriptions like "a bathroom with a sink, mirror, [...]" and "a kitchen with modern appliances and a countertop." **2. Spatial Consistency:** The example reveals spatial consistency within the generated trajectories. For instance, in Step 4, the agent correctly identifies the door with a potted plant, consistent with its position in Step 3. Such instances emphasize the preservation of spatial relationships across the generated trajectories. **3. Descriptive:** The generated trajectories incorporate a significant amount of captions and objects that do not directly relate to the given instruction, which plays a crucial role in preparing the agent to navigate successfully in real-world scenarios.

**Remark.** We cannot entirely rule out the possibility that the GPT-4 training set included the text instructions seen in R2R.[3] However, out of the 10,000 generated instructions, we did not find any instructions that were in the actual R2R dataset.

### 4.1.2 EXPERIMENTS

**Experimental setup.** We compare LangNav with the following baselines. **1. Random walk**, which selects a random action at each time step. **2. GPT-4 (Zero-shot / Few-shot):** We prompt GPT-4 to complete the trajectory by changing the task description of the template in § 3.2 (see appendix G for the full prompt). For the few-shot baseline, due to the context length we use one full navigation trajectory as a demonstration example. **3. NavGPT:** A contemporaneous work that also uses language space (via image captioning and object detection) to perform navigation, but purely with GPT-4 (Zhou et al., 2023a). Note that this is not strictly to comparable to our baselines as

---

[3]While the text instructions may have been encountered, the trajectories were unlikely to have been encountered during pretraining since we used vision systems to obtain the captions/objects.

**Instruction:**
Enter the hallway and follow it until you see a door with a potted plant nearby. Enter the room and locate the bookshelf. Stop next to the bookshelf.

**Step 1:**
To your **straight ahead** is, a bright living room with a comfortable sofa and a coffee table
To your **60 degree right** is, a kitchen with modern appliances and a countertop
To your **back** is, a dining area with a large table and chairs
To your **90 degree left** is, a hallway with framed pictures on the wall
You chose:
[a hallway with framed pictures on the wall]

**Step 2:**
To your **90 degree left** is, a cozy bedroom with a neatly made bed
To your **60 degree right** is, a bathroom with a sink, mirror, and a bathtub
To your **straight ahead** is, a long hallway with doors on both sides
To your **back** is, a living room with a sofa and a coffee table
You chose:
[a long hallway with doors on both sides]

**Step 3:**
To your **30 degree left** is, a door with a potted plant beside it
To your **90 degree left** is, a small office with a desk and a computer
To your **straight ahead** is, the end of the hallway with a window
To your **back** is, a hallway with picture frames on the wall
You chose:
[a door with a potted plant beside it]

**Step 4:**
To your **60 degree right** is, a large bookshelf filled with books and decorations
To your **back** is, a hallway with a potted plant by the door
To your **30 degree left** is, a cozy lounge area with a sofa and a coffee table
To your **90 degree left** is, a home office with a desk, chair, and a computer
You chose:
[a large bookshelf filled with books and decorations]

**Step 5:**
To your **90 degree right** is, a window with a view of the backyard
To your **straight ahead** is, a bookshelf with various items and books
To your **back** is, an entrance with a potted plant and a door leading to the hallway
To your **150 degree left** is, a comfortable lounge area with a sofa and a coffee table
You chose: **[Stop]**

Figure 3: An example of a generated trajectory from GPT-4. The example demonstrates a generated trajectory by following the pipeline in fig. 2, where an embodied agent follows the instruction and locates the bookshelf. See more examples in appendix H.

| Methods | # real | Val Seen | | | | Val Unseen | | | |
|---|---|---|---|---|---|---|---|---|---|
| | | NE↓ | OSR↑ | SR↑ | SPL↑ | NE↓ | OSR↑ | SR↑ | SPL↑ |
| Random Walk | 0 | 10.2 | 5 | 3 | 1 | 9.5 | 6 | 3 | 2 |
| GPT-4 (Zero-shot) | 0 | 10.5 | 15 | 9 | 8 | 10.2 | 17 | 10 | 8 |
| GPT-4 (Few-shot) | 1 | 10.1 | 17 | 10 | 9 | 9.9 | 22 | 13 | 11 |
| NavGPT* (Zhou et al., 2023a) | 0 | - | - | - | - | 6.5 | 42 | 34 | 29 |
| RecBert (Hong et al., 2021) | 10 | 10.8 | 9 | 7 | 6 | 10.1 | 13 | 9 | 9 |
| DuET (Chen et al., 2022) | 10 | 10.0 | 21 | 14 | 12 | 9.9 | 20 | 12 | 11 |
| LLaMA2-7B | 10 | 10.2 | 15 | 11 | 10 | 9.6 | 16 | 11 | 9 |
| LangNav (with LLaMA2-7B) | 10 | **7.5** | **39** | **31** | **27** | **7.0** | **42** | **32** | **28** |
| RecBert (Hong et al., 2021) | 100 | 9.3 | 27 | 20 | 19 | 9.4 | 26 | 19 | 17 |
| DuET (Chen et al., 2022) | 100 | 9.2 | 31 | 21 | 18 | 9.4 | 32 | 23 | 19 |
| LLaMA2-7B | 100 | 9.6 | 29 | 21 | 18 | 9.1 | 30 | 19 | 17 |
| LangNav (with LLaMA2-7B) | 100 | **7.4** | **40** | **32** | **28** | **7.1** | **45** | **34** | **29** |

Table 1: Results on the R2R dataset with 10 or 100 real world trajectories. Our LangNav approach finetunes LLaMA2-7B on the mixture of the real-world trajectories and 10,000 synthetic trajectories from GPT-4. *NavGPT relies on ground-truth distance information and is thus not strictly comparable to other baselines.

NavGPT assumes access to ground truth distance information. **4. RecBert**: a vision-based method that adopts a recurrent architecture proposed by Hong et al. (2021) to keep track of the trajectory history. **4. DuET**: another vision-based method which additionally builds representations of the global map during learning (Chen et al., 2022). **5. LLaMA2-7B**: a language-only baseline which does not make use of synthetically-generated data from GPT-4.

All finetuning methods use the same set of 10/100 trajectories. For these experiments we did not find significant differences in performance when using the object detection module, and hence we only rely on the image captioning system to give the language description of each view angle in the prompt template. See appendix A for the full training setup including hyperparameters.

**Results.** The results are shown in table 1. We find that GPT-4 zero- and few-shot results underperform the NavGPT baseline despite using the same backbone model, potentially due to NavGPT's use of chain-of-thought-style prompts (Wei et al., 2022; Kojima et al., 2023) as well as its use of ground truth distance information. Just finetuning LLaMA2-7B on the 10/100 gold trajectories does not perform well, although it is comparable to the vision-based policies. Training on a mixture of synthetic and gold trajectories improves performance by a nontrivial margin, and the LLaMA2-7B-based LangNav approaches the performance of NavGPT despite being many times smaller. (However our approach does require a small number of gold trajectories.) This indicates that our pipelined prompting strategy is an effective approach for distilling the rich navigation-relevant world knowledge within GPT-4 to a smaller (and more efficient) language model.

| # synthetic data | LLM | NE↓ | OSR↑ | SR↑ | SPL↑ |
|---|---|---|---|---|---|
| 2,000 | GPT-3.5 | 9.8 | 31 | 16 | 12 |
| 500 | GPT-4 | 8.0 | 38 | 25 | 21 |
| 2,000 | GPT-4 | 7.0 | 42 | 31 | 27 |
| 10,000 | GPT-4 | 7.0 | 42 | 32 | 28 |

Table 2: Performance on the Val Unseen set as we vary the number of synthetically generated data and the underlying LLM from which the synthetic data is generated.

| Methods | Pretraining Data | R2R data | Val Seen | | | | Val Unseen | | | |
|---|---|---|---|---|---|---|---|---|---|---|
| | | | NE↓ | OSR↑ | SR↑ | SPL↑ | NE↓ | OSR↑ | SR↑ | SPL↑ |
| RecBert | None | 10 | 10.8 | 9 | 7 | 6 | 10.1 | 13 | 9 | 9 |
| | | 100 | 9.3 | 27 | 20 | 19 | 9.4 | 26 | 19 | 17 |
| | ALFRED | 0 | 9.5 | 12 | 8 | 4 | 9.0 | 12 | 7 | 3 |
| | | 10 | 10.8 | 11 | 7 | 6 | 10.7 | 13 | 9 | 7 |
| | | 100 | 9.9 | 22 | 18 | 17 | 10.2 | 23 | 15 | 14 |
| LangNav | None | 10 | 10.3 | 17 | 10 | 8 | 9.8 | 20 | 11 | 8 |
| | | 100 | 9.0 | 25 | 20 | 18 | 9.2 | 25 | 17 | 15 |
| | ALFRED | 0 | 9.2 | 20 | 17 | 15 | 8.9 | 24 | 18 | 16 |
| | | 10 | 8.7 | 20 | 19 | 18 | 8.3 | 21 | 18 | 17 |
| | | 100 | 8.1 | 29 | 25 | 24 | 8.0 | 29 | 24 | 22 |

Table 3: Sim-to-real where we pretrain a navigation agent on the simulated ALFRED environment and finetune on the real-world R2R data. We use LLaMA-7B (Touvron et al., 2023a) as our backbone model, and compare against the RecBert (Hong et al., 2021) baseline.

We conduct an ablation study by varying both the number of synthetic trajectories and the source of synthetic data. As shown in table 2, increasing the number of synthetic trajectories generated by GPT-4 demonstrates a positive impact on performance, although the gains are marginal when going from 2,000 to 10,000 trajectories. Switching the synthetic data source from GPT-4 to GPT-3.5 results in a noticeable decline in performance, highlighting the necessity of using a strong backbone language models for generating synthetic data.

## 4.2 CASE STUDY 2: LANGUAGE AS A BRIDGE FOR SIM-TO-REAL TRANSFER

We next experiment with using language as a domain-invariant perceptual representation space to transfer a policy that has been trained on a simulated environment to the real-world R2R environment. We choose the popular ALFRED dataset (Shridhar et al., 2020) as our simulated environment. The ALFRED dataset, based on the AI2THOR environment (Kolve et al., 2017), provides language instructions for household tasks.

There are significant differences between ALFRED and R2R which makes straightforward sim-to-real transfer challenging. ALFRED uses images rendered from the synthetic AI2THOR environment, while R2R, based on the Matterport3D, incorporates images captured from real indoor environments. These image sources differ in texture, occlusion, illumination, and other visual aspects. ALFRED's navigation trajectories and instructions are also simpler and shorter compared to R2R's instructions. R2R instructions involve guiding the agent between rooms, whereas ALFRED trajectories mainly keep the agent within a single room. Finally in ALFRED, the agent is limited to rotating left/right by 90° and moving forward, while in R2R, the agent can move in any combination of 12 candidate heading directions and 3 elevation directions. See appendix B for further discussion of these differences, and see appendix A for the full experimental setup.

**Results.** We pretrain both RecBert (Hong et al., 2021) and LangNav on the simulated ALFRED environment and finetune on 0/10/100 R2R trajectories. LangNav uses LLaMA1-7b (Touvron et al., 2023a) as the language model. The evaluation results for both methods are presented in Table 3. Interestingly, for RecBert, pretraining on ALFRED actually *hurts* performance, potentially due to the model's overfitting to quirks of the simulated environment. And without any R2R data, RecBert performs near chance, whereas LangNav is able to exhibit some level of zero-shot transfer. Pretraining in ALFRED consistently leads to performance improvements for LangNav. This contrasting behavior between RecBert and LangNav highlights the potential of language as a domain-invariant perceptual representation for navigation.

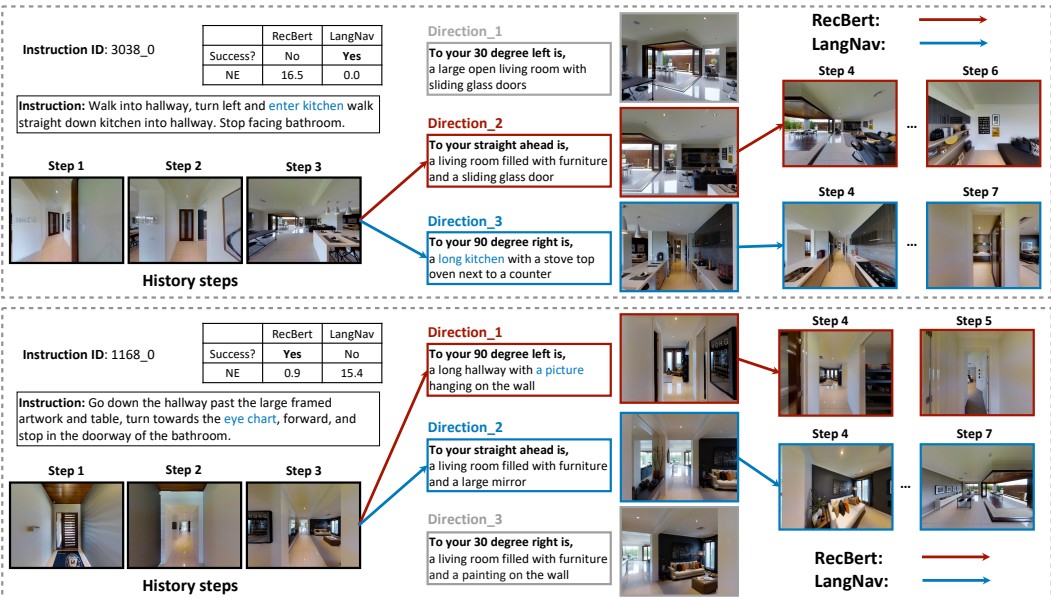

Figure 4: Qualitative results of comparing our LangNav and the vision-based method (RecBert Hong et al. (2021)). For each example, the chosen actions before the visualized step were identical so we put the history steps in the same row. The RecBert model is pretrained and fine-tuned on the full R2R train set, while our LangNav model is pre-trained on 2,000 GPT-4 synthetic trajectories and 100 real-world trajectories. NE: Navigation Error.

## 5 DISCUSSION

Here we discuss some qualitative results as well as limitations of our approach.

**Qualitative analysis.** We present two qualitative examples to illustrate the strengths and weaknesses of our approach when compared to the visual-based method shown in fig. 4. In the first example 3038_0, our LangNav agent successfully reaches the goal, whereas the vision-based RecBert fails to do so. The divergence between the two agents becomes evident at the third step when our LangNav agent correctly identifies the kitchen on the right and turns in that direction to enter it. In contrast, in the second example 1168_0, our LangNav agent falls short of reaching the goal due to a missed left turn at the third step. This discrepancy may be attributed to the agent's failure to perceive the eye chart on the left, which is not explicitly mentioned in the instruction's caption from the left direction. These two instances highlight the proficiency of our LangNav agent in grounding observed concepts within the navigation instruction. However, it also underscores a potential limitation where certain crucial visual concepts may not be adequately represented in the language representations.

**Limitations.** While we find that LangNav is promising in settings where only a handful of real trajectories are available, on the full dataset it still underperforms vision-based agents by a nontrivial margin, as shown in table 2. This is especially true when compared to state-of-the-art approaches such as ScaleVLN which make use of large-scale pretraining data as well as more involved imitation/reinforcement learning algorithms that require access to an environment oracle. However, we note that while LangNav underperforms baselines in data-rich regimes, it overfits less to scenes seen during training, as demonstrated by the smaller drop in performance when applying the policy to unseen scenes during training.

Language naturally abstracts away low-level perceptual details which we find to be beneficial for efficient data generation and sim-to-real transfer. However, this is also a serious limitation insofar as a picture really *is* worth a "thousand words" in some cases. Our paper should be seen as more of an exploratory exercise to test the potential of language as a perceptual representation for navigation (which has been understudied compared to use of language models in other embodied tasks) rather than a serious attempt at the state-of-the-art. We are certainly not suggesting the abandonment of traditional (continuous) vision features for vision-language navigation. An interesting direction might involve the use of both vision- and language-based perceptual representations for navigation.

| Methods | Training data | Needs Oracle | Val Seen | Val Unseen | Drop |
|---|---|---|---|---|---|
| Seq2Seq (SF) Anderson et al. (2018b) | R2R | No | 38.6 | 21.8 | 16.8 |
| RCM Wang et al. (2019) | R2R | Yes | 67.4 | 42.5 | 24.9 |
| Speaker-Follower Fried et al. (2018) | R2R+SpeakerAug. | Yes | 70.1 | 54.6 | 15.5 |
| RecBert[†] Hong et al. (2021) | R2R+PREV | Yes | 71.8 | 54.5 | 17.3 |
| HAMT Chen et al. (2021b) | R2R+PREV | Yes | 75.0 | 65.7 | 9.3 |
| ScaleVLN Wang et al. (2023) | R2R+PREV | No | 67.2 | 47.4 | 19.8 |
| ScaleVLN Wang et al. (2023) | R2R+PREV | Yes | 76.9 | 72.9 | 4.0 |
| ScaleVLN Wang et al. (2023) | R2R+PREV+ScaleVLN | No | 71.1 | 57.0 | 14.1 |
| ScaleVLN Wang et al. (2023) | R2R+PREV+ScaleVLN | Yes | 80.5 | 78.1 | 2.4 |
| LangNav | R2R | No | 55.0 | 43.2 | 11.8 |
| LangNav (M) | R2R+ALFRED | No | 55.9 | 45.6 | 10.3 |

Table 4: Comparison with state-of-the-art vision-based methods on the R2R dataset when trained on the full dataset. We use success rate (SR) as the performance metric. "Needs oracle" indicates that the model needs to rely on an oracle during training that can give the ground-truth next action based on a sampled path from the model. Reimplemented without pretraining on the val_unseen set. (M): Multi-Task model, see appendix C for details.

## 6 RELATED WORK

**Language Models for Task Planning.** Several studies have explored language-based planning Jansen (2020); Sharma et al. (2021); Li et al. (2022b); Huang et al. (2022a); Ahn et al. (2022); Huang et al. (2022b). Huang et al. (2022a) use GPT-3 Brown et al. (2020) and Codex Chen et al. (2021a) for action plan generation with semantic translation using Sentence-RoBERTa Huang et al. (2022a). SayCan Ahn et al. (2022) grounds actions using FLAN Wei et al. (2021) and action value functions Shah et al. (2021). Huang et al. (2022b) explore incorporating grounded feedback into LLMs, while Xiang et al. (2023) propose enhancing LLMs' with embodied task instructions.

**Instruction Tuning.** FLAN Wei et al. (2021) demonstrated the effectiveness of fine-tuning LLMs with instructions from multiple tasks. Instruction tuning has been widely applied to prominent large language models, including InstructGPT Ouyang et al. (2022), FLAN-T5 Chung et al. (2022), FLAN-PaLM Chung et al. (2022), and OPT-IML Iyer et al. (2022), but mainly focused on traditional language tasks. Our work instead finetunes LLMs for embodied navigation tasks using language descriptions of perceptual representations. There has been much recent work finetuning smaller language models such as LLaMA on synthetic instruction-following data generated by GPT-3.5/GPT-4 (Peng et al., 2023; Taori et al., 2023; Chiang et al., 2023; Wu et al., 2023). For example, LaMini-LM (Wu et al., 2023) generates synthetic instructions and then employs GPT-3.5 for generating the response. Our method differs from those as we focus on using GPT-4 to generate synthetic navigation trajectories, which to our knowledge has not been investigated before.

**Embodied Vision-and-Language Navigation.** The vision and language navigation task has gained attention since its introduction Anderson et al. (2018a) with the R2R dataset. Approaches such as the speaker-follower model Fried et al. (2018) and environmental dropout method Tan et al. (2019) improve generalization. Reinforced cross-modal matching Wang et al. (2019) and self-monitoring Ma et al. (2019) enhance performance. Hong et al. Hong et al. (2020) propose a language and visual entity relation graph. Recent advancements include VLBERT-based methods Hong et al. (2021) and object-informed sequential BERT Qi et al. (2021). Qiao et al. Qiao et al. (2022) incorporate additional pretext tasks into VLN pre-training based on Hong et al. (2021). ALFRED Shridhar et al. (2020) involves interactive actions in a synthetic environment Kolve et al. (2017), with methods utilizing dense single vector representations Shridhar et al. (2020); Singh et al. (2021); Pashevich et al. (2021); Kim et al. (2021); Blukis et al. (2022) or a panoramic view space Suglia et al. (2021). In contrast, our method distinguishes itself by operating solely on language input, as our blind navigation agent doesn't rely on vision-based features.

## 7 CONCLUSION

We show that we can learn to navigate in a realistic environment by using language to (1) easily generate synthetic trajectories and (2) transfer knowledge from a simulated environment. Our work demonstrates the potential of language to serve as a domain-invariant perceptual representation for egocentric navigation in low-data regimes with only a handful of real-word trajectories.

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

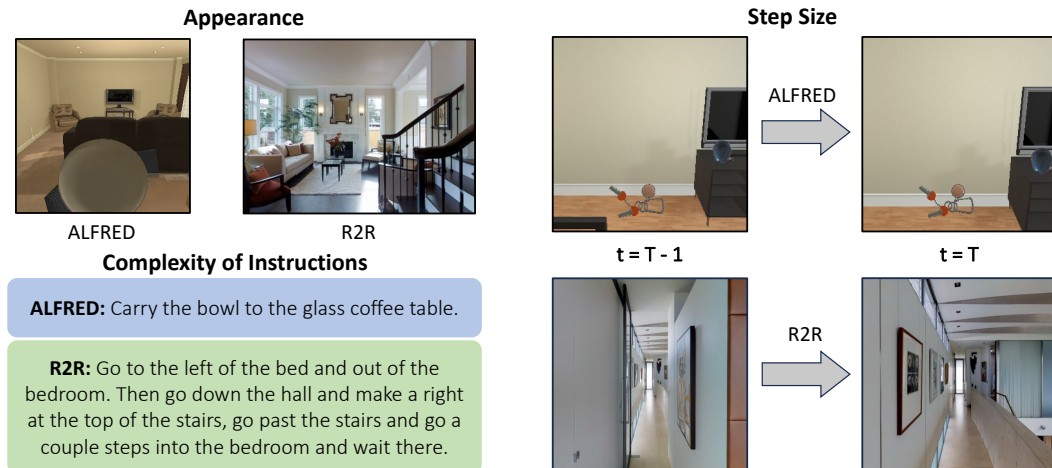

Figure 5: Task gap between ALFRED and R2R. We highlight notable distinctions between the navigation tasks in ALFRED and R2R, encompassing variations in appearance, step size, and instruction complexity. See appendix B for more details.

## A IMPLEMENTATIONS DETAILS

We used the LLaMA-7B model Touvron et al. (2023a) and the LLaMA2-7B model Touvron et al. (2023b) for our method, fine-tuning it on 72 V100-32GB GPUs with a batch size of 144. The training tokens had a maximum length of 1024, while during inference, the maximum length was set to 2048. The AdamW optimizer Loshchilov & Hutter (2017) with a learning rate of $2 \times 10^{-5}$ and weight decay of 0 was employed for optimization. The WarmupDecayLR learning rate scheduler was used for learning rate scheduling. For image captioning in both the R2R and ALFRED tasks, BLIP Li et al. (2022a) was utilized. Deformable DETR Zhu et al. (2020) was used for object detection in the R2R dataset, with suppression of outdoor object categories. We used the ground-truth object detection results provided in ALFRED when we generated the instruction-following pairs in § 4.2. When prompting GPT-4 API, we set the temperature as 1 and top_p as 1. The cost of collecting the generated trajectories by prompting GPT-4 API OpenAI (2023) was around $500. In the few-shot learning experiments in § 4.1 and § 4.2, we set $\rho = 0$. While when fine-tuning with the full train set in § 5, we set $\rho = 0.2$. We pretrain on 128K ALFRED instruction-following pairs whose format is given in § 3.2. We augment the observations in ALFRED to 12 views and randomly mask a variable number of views to mimic the irregular number of candidates in R2R.

## B DIFFERENCES BETWEEN ALFRED AND R2R.

There are significant differences between ALFRED and R2R which makes straightforward sim2real transfer challenging. These differences include:

**Visual appearance.** ALFRED uses images rendered from the synthetic AI2THOR environment, while R2R, based on the Matterport3D, incorporates images captured from real indoor environments. These image sources differ in texture, occlusion, illumination, and other visual aspects.

**Step size.** There is a difference in step sizes between the two tasks (see the right part of fig. 5). ALFRED uses a step size of 0.25 meters, while R2R has larger and more variable step sizes. To bridge this gap, we consolidate four consecutive MoveAhead steps into a single step along the ALFRED trajectory.

**Action type.** A complete ALFRED trajectory includes not only navigation actions but also interaction actions, where the interaction actions are combined with a target object to change the state of the surrounding environment. In order to filter the interaction actions in ALFRED, we divide each ALFRED trajectory into multiple sub-trajectories and keep the sub-trajectories that are labeled with the GotoLocation tag.

Table 5: Performance of the Multi-task Model on R2R. We demonstrate the multi-task capability of the LM agent. For single-task models, each model is trained within the task data. We trained the multi-task model with data from both R2R and ALFRED tasks.

| Models | R2R Seen | | R2R Unseen | |
|---|---|---|---|---|
| | SR↑ | SPL↑ | SR↑ | SPL↑ |
| Single-Task | 55.0 | 51.0 | 43.2 | 37.9 |
| Multi-Task | **55.9** | **51.7** | **45.6** | **40.0** |

Table 6: Performance of the Multi-task Model on ALFRED. ST: Single-Task. MT: Multi-Task.

| | ALFRED Seen | | ALFRED Unseen | |
|---|---|---|---|---|
| | Task↑ | GC↑ | Task↑ | GC↑ |
| ST | 0.0 (0.0) | 6.0 (4.7) | 0.5 (0.1) | 9.5 (7.8) |
| MT | 0.0 (0.0) | **6.4 (5.0)** | **0.6 (0.2)** | **9.8 (7.8)** |

**Instruction complexity.** Due to trajectory splitting, ALFRED's navigation trajectories and instructions appear simpler and shorter compared to R2R's instructions. R2R instructions involve guiding the agent between rooms, whereas ALFRED trajectories mainly keep the agent within a single room.

**Action space.** In ALFRED, the agent is limited to rotating left/right by $90°$ and moving forward, while in R2R, the agent can move in any combination of 12 candidate heading directions and 3 elevation directions. The number of available movement directions is irregular. This difference in action space makes R2R trajectories more human-like. To address this, we introduce randomness by adding or reducing a heading offset of $±30°$ to the agent's direction at each step in ALFRED, allowing rotations of $30°$ or $60°$ in addition to $90°$.

## C  MULTI-TASK PERFORMANCE

One of the advantages of our approach is its inherent suitability for multitasking. Similar to LLMs use instruction to handle multiple language tasks concurrently, we consolidate task information and inputs into instructions. To validate the multitasking capability of our method, we extend its application to the ALFRED task.

**Metrics on ALFRED.**  We evaluate our model on ALFRED using two metrics: *Task Success* (Task) and *Goal-Condition Success* (GC). Task Success measures the ratio of trajectories where object positions and state changes accurately match all task goal conditions at the end. GC assesses the ratio of completed goal conditions in each action sequence. Task Success is only considered successful when GC is also 1. On average, each ALFRED task has 2.55 goal conditions. We also calculate the *Path Length Weighted Metrics* (PLW) for both Task and GC, which normalize the metrics based on the actual action sequence length.

**Results of the Multi-Task Model.**  In ALFRED task, we set $\rho = 0$ as the expert policy in ALFRED is suboptimal. To save training time and balance the data amount between R2R and ALFRED, we utilize only 50% of the training dataset, resulting in a dataset for ALFRED with 386K data pairs. For R2R task training, we maintain $\rho = 0.2$ and run each demonstration trajectory twice, resulting in a training set size of 235K for R2R. Consequently, the merged dataset for the multitask model contains a total of 621K instruction-following data pairs. We select VLN Bert Hong et al. (2021) as the baseline for the R2R task and Seq2seq model Shridhar et al. (2020) for the ALFRED task. Given the substantial differences between the R2R task and the ALFRED task (§ 4.2), our method is, to the best of our knowledge, the first model that simultaneously addresses these two tasks. In table 5 and table 6, we find that the multitask model exhibits superior performance compared to the single-task models. These results underscore the capability of our method to effectively handle multiple highly diverse tasks.

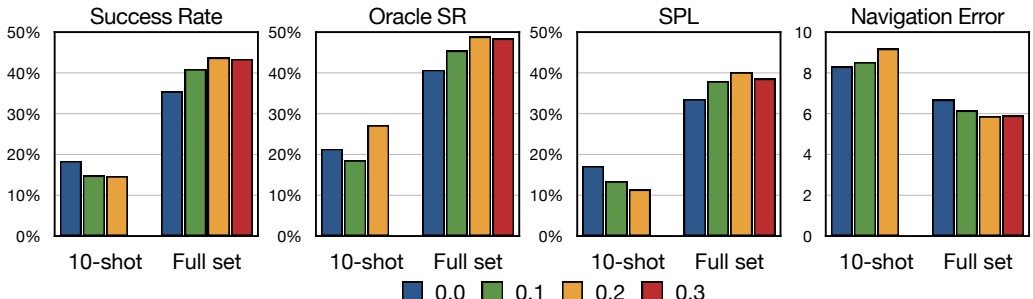

Figure 6: Investigating the Impact of the Randomness Factor $\rho$ on Model Performance. This image caption depicts an ablation study exploring the influence of the randomness factor $\rho$ on our model's performance in both few-shot learning and full-set training scenarios. We test $\rho$ with values of 0.0, 0.1, 0.2, and 0.3.

## D IMPACT OF THE RANDOMNESS FACTOR

We conduct the ablation study to investigate the impact of the randomness factor $\rho$ on the model's performance in both few-shot learning and full-set fine-tuning scenarios. Interestingly, we observe different behaviors of the model with varying $\rho$ values in these scenarios. Fig. 6 illustrates our findings. In the 10-shot scenario, increasing $\rho$ negatively affected the model's performance. However, in the full-set scenario, the model achieved optimal performance at $\rho = 0.2$. We propose a metaphorical explanation: for effective few-shot learning, the language model behaves like an infant, relying on highly accurate demonstrations, while for large-scale imitation learning, the language model behaves like an adult, benefitting from occasional detours (introduction of randomness factor $\rho$) to enhance robustness.

## E EXAMPLE OF THE FULL TRAJECTORY

We plot an example of a full text-based trajectory in R2R as we mentioned in § 3.2 as bellow:

```
You are a navigation agent who must navigate according to
instructions given only descriptions of your current position via
natural language.  The natural language description is sometimes
incorrect.

### Instruction:
Go across the kitchen passed the pizza oven into the next room.
Stop just inside the bedroom.

### Trajectory:
Step 1:

To your 60 degree left is,
a kitchen with a stove, sink, and refrigerator
Details: oven, bowl, dining table, microwave, refrigerator

To your 30 degree left is,
a kitchen with white cabinets and a white stove top oven
Details: oven, bowl, microwave, potted plant, chair

To your 60 degree right is,
a living room filled with furniture and a fire place
Details: couch, chair

To your back is,
```

```
a room with a bed and a painting on the wall

You chose:
a living room filled with furniture and a fire place

Step 2:

To your 150 degree left is,
a kitchen with a stove top oven sitting next to a counter
Details: oven, dining table, microwave, refrigerator, chair

To your back is,
a kitchen with a wooden table and white appliances
Details: oven, bowl, dining table, microwave, refrigerator, apple,
couch, chair

To your 30 degree right is,
a living room with a couch a chair and a window
Details: couch

To your 120 degree right is,
a dining room table with a bowl of fruit on it
Details: chair, bowl, dining table

To your 150 degree right is,
a bowl of fruit sits on a wooden table
Details: couch, chair, bowl, dining table

You chose:
a living room with a couch a chair and a window

Step 3:

To your back is,
a kitchen with a table, chairs, and stairs
Details: oven, dining table, refrigerator, potted plant, chair

To your 150 degree right is,
a room with a table, chairs, and stairs
Details: oven, chair, refrigerator, dining table

To your straight ahead and 30 degree down is,
a view of a hallway from the top of stairs
Details: refrigerator

To your 90 degree right and 30 degree up is,
a room with a staircase and a mirror on the wall
Details: toilet

You chose:
a view of a hallway from the top of stairs

Step 4:

To your back and 60 degree up is,
a living room filled with furniture and a ceiling fan
Details: oven, potted plant, refrigerator

To your 90 degree left and 30 degree up is,
```

```
a living room with a couch and a table

To your straight ahead and 30 degree up is,
a bedroom with a bed and a mirror on the wall
Details: bed

You chose:
a bedroom with a bed and a mirror on the wall

Step 5:

To your back is,
a hallway leading to a kitchen and living room
Details: refrigerator, potted plant

To your 30 degree left is,
a room with a wooden door and a mirror

To your straight ahead is,
a bedroom with a bed, dresser, mirror and a ceiling fan
Details: potted plant, bed

To your 30 degree right is,
a bedroom with a bed and a ceiling fan
Details: potted plant, bed

To your 60 degree right is,
a bedroom with a bed, dresser and mirror
Details: potted plant, bed

You chose:
stop
```

## F  COMPLETE PROMPT TEMPLATE OF GENERATING TRAJECTORIES FOR GPT-4

We list our complete templates for prompting GPT-4 to generate synthetic instructions (Phase I) and synthetic trajectories to fulfill the instruction (Phase II).

**Phase I:**  The template of phase I is listed as follows:

```
I am going to give you example instructions written by humans to
train a deep learning-based navigation agent acting inside a home.
These example instructions are intended to be completed by the
navigation agent in 5-7 steps.
- {real_instruction_1}
- {real_instruction_2}
- {real_instruction_3}
Your goal is to write 10 more instructions like the above that can be
used to train a navigation agent. Since the navigation agent will be
navigating in different home environments, your instructions should
also be diverse and cover a wide range of home environments and
rooms. You should make sure that the instruction can be completed
by an agent in 5 to 7 steps.
```

**Phase II:**  The template of phase II is listed as follows:

```
Here is an example of a large language model acting as a
blind navigation agent in an indoor environment through text
descriptions. The agent is given an instruction at the start and
must follow the instruction.  At each time step, the agent is
given descriptions of its field of view via the following template:

To your [VIEW] is [CAPTION]
- [VIEW] consists of the agent's visible field of view (e.g., 30
degrees right, 120 degrees left, etc.)
- [CAPTION] is the text description of that view obtained from an
image captioning model

#Example 1
### Instruction: {real_instruction_example}
### Trajectory: {real_trajectory_example}

Now I will give you another instruction.  Please generate a
trajectory of 5-7 steps that would complete the instruction.
#Example 2
### Instruction: {synthetic_instruction}
```

## G   PROMPTS OF ZERO-SHOT AND FEW-SHOT NAVIGATION FOR GPT-4

Here we attach the the task description $D$ in the prompt template for prompting GPT-4 to navigate in the R2R evaluation dataset.

Zero-shot:

```
You are a navigation agent who must navigate according to
instructions given only descriptions of your current position via
natural language.  The natural language description is sometimes
incorrect.

At each step, you will be given several directions and captions
for each direction. You must choose one direction by printing only
the [caption_of_the_direction] or choose "Stop" if you think the
goal is reached.

For example:

Input:

To your [direction_1] is, [caption of the direction_1].
......
To your [direction_N] is, [caption of the direction_N].

You choose:

Output: [caption of the direction_3]

Hint:  You should use the information inside the instructions,
history steps, and current observations to make the decision.
```

Few-shot:

```
You are a navigation agent who must navigate according to
instructions given only descriptions of your current position via
natural language.  The natural language description is sometimes
incorrect.
```

At each step, you will be given several directions and captions
for each direction. You must choose one direction by printing only
the [caption_of_the_direction] or choose "Stop" if you think the
goal is reached.

For example:

Input:

To your [direction_1] is, [caption of the direction_1].
......
To your [direction_N] is, [caption of the direction_N].

You choose:

Output: [caption of the direction_3]

And here is an example trajectory:

### Instruction:
Go down the stairs. Turn right and go down the hallway. Turn right
and stand near the fireplace.

### Trajectory:
Step 1:

To your straight ahead is,
an ornate doorway leading to another room

To your 60 degree right is,
a red carpeted staircase leading to a chandelier

To your 120 degree right is,
a room with a red carpet and a large mirror

To your back and 30 degree down is,
a room with a red carpet and two windows

To your 120 degree left is,
a room with a red carpet and gold trim

You chose:
a room with a red carpet and gold trim

Step 2:

To your 150 degree right is,
a very ornate staircase in a house with red and white striped chairs

To your back is,
a red carpeted hallway leading to a staircase

To your 150 degree left is,
a hallway with a red carpet and a chandelier

To your 120 degree left is,
a room with a red carpet and a chandelier

```
To your 90 degree left is,
a room with a chandelier and two windows

To your 60 degree left is,
a room with a red carpet and a large mirror

To your 30 degree right is,
a hallway with a red carpet and wooden doors

You chose:
a hallway with a red carpet and wooden doors

Step 3:

To your back is,
a hallway with a red carpet and a chandelier

To your straight ahead is,
a hallway with a red carpet and a gold ceiling
a hallway with a red carpet and a gold ceiling

You chose:
a hallway with a red carpet and a gold ceiling

Step 4:

To your 90 degree right is,
a living room with a chandelier and a fireplace

To your 120 degree right is,
a room with a fireplace and a chandelier in it

To your back is,
a hallway with a red carpet and gold trim

To your 90 degree left is,
a room with a chandelier and a table in it

To your 30 degree right is,
a living room with a chandelier and a couch

You chose:
a living room with a chandelier and a fireplace

Step 5:

To your straight ahead is,
a living room filled with furniture and a fire place

To your 30 degree left is,
a living room filled with furniture and a chandelier

To your 150 degree right is,
a room with a chandelier and a table in it

To your back is,
a hallway with a chandelier and a mirror on the wall

To your 90 degree left is,
```

```
a living room filled with furniture and a chandelier

You chose:

Stop

Hint:  You should use the information inside the instructions,
history steps, and current observations to make the decision.

Now let's start!
```

## H  MORE EXAMPLES OF GENERATED TRAJECTORIES

We plot more examples of the generated trajectories from GPT-4 below:

```
Example 1:

        ### Instruction
        Begin in the basement, walk towards the pool table. Pass the pool
        table and ascend the staircase. At the top of the stairs, turn
        right and walk through the hallway. Turn left into the den, walk
        through and stop by the window. ### Trajectory:

        Step 1:

        To your straight ahead is,
        a basement with a pool table, a bar, and a seating area

        To your 90 degree left is,
        a wall with various framed photos and a door

        To your 150 degree right is,
        a storage area filled with boxes and shelves

        You chose:
        a basement with a pool table, a bar, and a seating area

        Step 2:

        To your back is,
        a basement entrance with a staircase leading down

        To your 120 degree left is,
        a basement with a bar, television, and several chairs

        To your straight ahead is,
        a basement with a pool table, a dartboard, and a bar

        To your 90 degree right is,
        a staircase leading upwards with a wooden railing

        You chose:
        a basement with a pool table, a dartboard, and a bar

        Step 3:

        To your back is,
        a basement with a seating area and framed photos on the wall
```

```
To your straight ahead is,
a pool table surrounded by chairs and a dartboard on the wall

To your 150 degree left is,
a basement with a bar, television, and several chairs

To your 90 degree right is,
a staircase leading upwards with a wooden railing

You chose:
a staircase leading upwards with a wooden railing

Step 4:

To your back is,
a basement with a pool table and a dartboard on the wall

To your 90 degree left is,
a hallway with various framed photos on the walls

To your straight ahead is,
a den with a large bookshelf, couch, and coffee table

You chose:
a hallway with various framed photos on the walls

Step 5:

To your back is,
a staircase leading downwards into the basement

To your straight ahead is,
a long hallway with several doors leading to other rooms

To your 120 degree right is,
a den with a comfortable seating area and a bookshelf

You chose:
a long hallway with several doors leading to other rooms

Step 6:

To your back is,
a hallway with various framed photos on the walls

To your 90 degree left is,
a den filled with cozy furniture, bookshelves, and a television

To your 90 degree right is,
a bedroom with a large bed, dresser, and a window

You chose:
a den filled with cozy furniture, bookshelves, and a television

Step 7:

To your back is,
a hallway leading to other parts of the house
```

```
        To your 90 degree left is,
        a den with several bookshelves, a fireplace, and a large rug

        To your straight ahead is,
        a window overlooking the garden outside

        You chose:
        Stop

Example 2:

        ### Instruction
        From the garage entrance, walk into the kitchen. Turn left and
        walk past the stove. Continue straight into the family room, then
        walk towards the sliding glass doors. Step outside and stop near
        the barbecue grill.

        ### Trajectory:
        Step 1:

        To your straight ahead is,
        a clean garage with a parked car and a workbench

        To your 90 degree left is,
        a laundry room with a washer and a dryer

        To your back is,
        a driveway leading up to a garage door

        To your 120 degree left is,
        a garage entrance with an open door leading inside the house

        You chose:
        a garage entrance with an open door leading inside the house

        Step 2:

        To your 30 degree left is,
        a kitchen with a stove, a refrigerator, and a microwave oven

        To your 90 degree right is,
        a small mudroom with a shoe rack and a coat hook

        To your back is,
        a view of a garage with a car and a workbench

        You chose:
        a kitchen with a stove, a refrigerator, and a microwave oven

        Step 3:

        To your straight ahead is,
        a large kitchen with a stove and a range hood

        To your 30 degree right is,
        a kitchen with a center island and a sink on the countertop

        To your 150 degree left is,
```

a kitchen with a refrigerator and a microwave oven

To your 120 degree right is,
a tiled floor leading to a hallway with a picture on the wall

You chose:
a large kitchen with a stove and a range hood

Step 4:

To your straight ahead is,
a cozy family room with a TV and a fireplace

To your 90 degree right is,
a kitchen with a sink and a dishwasher

To your back is,
a kitchen with a countertop, a cooktop, and an oven

You chose:
a cozy family room with a TV and a fireplace

Step 5:

To your 30 degree left is,
a sliding glass door leading to the backyard

To your 90 degree right is,
a comfortable sofa with a coffee table in front of a TV

To your back is,
a large kitchen with a stove and a range hood

You chose:
a sliding glass door leading to the backyard

Step 6:

To your straight ahead is,
a well-maintained backyard with a barbecue grill and a dining table

To your 90 degree left is,
a sliding glass door leading inside the family room

To your back is,
a cozy family room with a TV and a fireplace

You chose:
a well-maintained backyard with a barbecue grill and a dining table

Step 7:

To your 90 degree left is,
a barbecue grill on a patio in the backyard

To your straight ahead is,
a backyard with a dining table and chairs

To your 120 degree right is,

```
        a backyard with a garden and a swing set

        To your back is,
        a sliding glass door leading inside the family room

        You chose:
        Stop

Example 3:

        ### Instruction
        Stand in front of the fireplace in the living room. Walk diagonally
        across the room, passing the sofa. Proceed through the door into
        the hallway. Turn left and continue toward the guest room. Stop
        just before entering and wait.

        ### Trajectory:
        Step 1:

        To your straight ahead is,
        a living room with a fireplace and two chairs

        To your 30 degree left is,
        a living room with a large window and a dark brown couch

        To your 90 degree right is,
        a living room with a flat screen tv and a view of the backyard

        You chose:
        a living room with a fireplace and two chairs

        Step 2:

        To your 60 degree left is,
        a living room with a fireplace and a floor lamp

        To your 30 degree right is,
        a living room with a dark brown couch and a coffee table

        To your straight ahead is,
        a living room with a white rug in the middle

        You chose:
        a living room with a dark brown couch and a coffee table

        Step 3:

        To your back is,
        a living room with a fireplace and a christmas tree

        To your 150 degree left is,
        a living room with two chairs and a painting on the wall

        To your straight ahead is,
        a wooden door leading to a hallway

        You chose:
        a wooden door leading to a hallway
```

```
Step 4:

To your 90 degree left is,
a hallway with a view of a staircase and a table

To your straight ahead is,
a hallway with a painting on the wall and an open door

You chose:
a hallway with a painting on the wall and an open door

Step 5:

To your back is,
a hallway with a wooden floor and a closed door

To your 120 degree left is,
a guest bedroom with a neatly made bed and a dresser

To your 30 degree right is,
a hallway with white walls and floor-to-ceiling mirrors

You chose:
Stop just before entering the guest bedroom
```

