# OpenReview forum: "LangNav: Language as a Perceptual Representation for Navigation"
_ICLR.cc/2024/Conference — ICLR 2024 Conference Withdrawn Submission_

### Official Review · Reviewer_MJeR · 2023-10-30

**Soundness:** 4 excellent
**Presentation:** 4 excellent
**Contribution:** 2 fair
**Rating:** 6
**Confidence:** 4

**Summary:**

The objective of this article is to leverage language as a perceptual representation for vision-and-language navigation. Their system uses off-the-shelf vision systems (for image captioning and object detection) to convert an agent's egocentric panoramic view at each time step into natural language descriptions.

**Strengths:**

While it has become a common practice nowadays, leveraging the remarkable understanding and reasoning capabilities of large language models as opposed to vision models, makes using language as an intermediate representation is the optimal choice. The system efficiently achieves competitive results while utilizing minimal resources. The article effectively presents its ideas, provides sufficient experimental evidence to support its viewpoint, and offers a detailed implementation statement.

**Weaknesses:**

While it is indeed a valid approach to utilize language as a perceptual representation for vision-and-language navigation, there remains a significant domain gap in accurately converting vision into language. This discrepancy can result in an imprecise representation of language, subsequently impacting understanding and reasoning processes. The author should delve into more comprehensive discussions and conduct experiments to explore the influence of various vision models and propose potential solutions for rectifying inaccuracies when the vision model fails.

**Questions:**

Could you please provide a comparison between the latest COT technology on GPT-4 in terms of language representation (for instance, Graph of Thoughts: Solving Elaborate Problems with Large Language Models, not just NavGPT) and the fine-tuned LangNav to validate the effectiveness of fine-tuning LLaMA with synthetic data? It is important to highlight the advantages of such an approach, as the allocation of resources without clear benefits would be deemed unnecessary.

---

> ### Author Response · Authors · 2023-11-22
> **Response**
>
> Thanks for the review!
>
> **Gap in converting vision into language**
>
> We agree! To see whether (usual) continuous vision features can rectify close this gap, we conducted additional experiments where we augment a traditional vision-based navigation model with language features. More specifically, the original RecBert uses ResNet-152 to extract the visual feature to represent the image, while our extension additionally concatenates the language feature of the caption to be the image representation. The captions are the same as what are used in LangNav and the features are extracted by BERT-base. We pre-train and fine-tune the RecBert and its extension with 100-shot training trajectories and full R2R train set. The results are listed in the tables below.
>
>  | eval env   | train trajectories | perceptual features | SR↑   | SPL↑  |
> |------------|:--------------------:|:----------------------:|------|------|
> |  val_seen  |          100         |           vision   only         | 19.9 | 18.6 |
> | val_unseen |          100         |           vision  only         | 19.0 | 17.4 |
> |  val_seen  |          100         |           vision + language       |**22.8**|**21.5**|
> | val_unseen |          100         |           vision + language        |**19.3**|**18.0**|
>
> | eval env   | train trajectories  | perceptual features | SR↑   | SPL↑  |
> |------------|:--------------------:|:----------------------:|------|------|
> |  val_seen  |      full train      |            vision  only           | 58.5 | 55.4 |
> | val_unseen |      full train      |            vision  only           | 47.1 | 43.4 |
> |  val_seen  |      full train      |           vision + language        |**61.4**|**57.1**|
> | val_unseen |      full train      |           vision + language        |**48.8**|**44.1**|
>
> We can see that the language features improve the performance in both few-shot and full learning cases, which indicates that language-based features can help the visual perception module for the VLN agent. We hope that this addresses the reviewer's concerns the domain gap that may result in converting vision into language.
>
> **Q1: Could you please provide a comparison between the latest COT technology on GPT-4 in terms of language representation (for instance, Graph of Thoughts: Solving Elaborate Problems with Large Language Models, not just NavGPT)...**
>
> Thanks for the question! Incorporating more sophisticated reasoning techniques such as Graph of thought and tree of thought is an interesting suggestion. However, we note that it is not at all clear how one would apply these more sophisticated prompting techniques to the navigation case (as of now). Note that  NavGPT already uses chain-of-thought, and hence we do compare against a strong COT-based baseline. We will discuss this more in the next version of the paper.

---

> ### Comment · Reviewer_MJeR · 2023-11-22
> **Final Rating**
>
> Thanks to the authors for responding to my comments. After reading all the reviews and responses, I have understood the concerns of other reviewers and the authors' detailed rebuttals.
>
> Here, I would like to succinctly summarize the main strengths and weaknesses of this paper:
>
> **Strengths:**
> - Extensive comparisons across recent VLN baselines. However, during the rebuttal period, one experiment was conducted to address the concerns of five reviewers, although our concerns varied.
> - Language as a perceptual representation for VLN. While I agree with the weakness 1 of Reviewer PNyi, which the authors have rebutted and made commitments. The motivation is sound, nonetheless, the latest work in Vision-Language Models (VLM), even the powerful GPT-4V, still suffers from significant hallucination issues. Therefore, I wish to see more analysis and discussion on how to correct language representation failures, rather than always relying on an erroneous, discrete, and hallucinatory representation. I'm not questioning your argument.
>     > ... language-based features can help the visual perception module for the VLN agent.
>
> - I concur with this point of Reviewer z42V, *i.e.*, *"a good paper to motivate future work in this direction"*. Besides this point, I agree with the authors' rebuttal and strength 1 of Reviewer 2DEF.
>     > - Given that no new models or methods are developed, the key contributions of this paper are relatively light.
>     > - Overall, I think this is a good workshop paper to motivate future work in this direction, but in itself, does not meet the bar as a conference paper.
>
> **Weaknesses:**
> - The biggest weakness of this paper, I agree with the weakness 4 of Reviewer 2DEF.
>     > I believe one of the most important reasons for introducing LLMs to embodied AI tasks is to leverage LLMs' exceptional language understanding, task planning, tool-using, explicit reasoning with commonsense knowledge, communication, etc., abilities...
>
> I still believe that it is necessary to explore and analyze it (as mentioned in strength 2) in the future. I expect it to get better language perceptual representations, rather than directly applying it to navigation task.
>
> Overall, this paper can stimulate discussion and bring new ideas to the VLN community, supported by substantial work. I decided to maintain my rating as "6: marginally above the acceptance threshold".

---

> > ### Author Response · Authors · 2023-11-22
> > **More examples of LangNav's ability**
> >
> > We want to thank the reviewer for maintaining the rating after reading all the other reviews and our responses!
> >
> > To show more about the ability of LangNav, we show three more examples attached in [LangNav's Navigation](https://docs.google.com/document/d/e/2PACX-1vTVmWvp1gHwLcc8ZY8ybMZ-6uDannhGZ4pBn2GO2lxwiE8fW_zTcJi-mPGxaWzz3BP1MB0vvEu0T-_T/pub).
> >
> > In **Example #1**, we have observed that, although the instruction of where to stop is vague and ambiguous ("Wait at the entrance."), our LangNav agent manages to stop in front of the double wooden doors which is the most likely to be the entrance.
> >
> > In **Example #2**, from Step 2 to Step 4, our LangNav agent erroneously enters a bathroom on the right in Step 2, but then corrects its course by leaving the bathroom and proceeding to the living room. In contrast, we have also visualized the trajectory (Example #2) of RecBert under the same instructions in [RecBert's Navigation](https://docs.google.com/document/d/e/2PACX-1vSDufWgGM-0r60U1dtxg7DfeF-uEsI_z2WIkK9UyyBp2RG2zIXpgYpAERFDtDchUphXb00_Gihzi7sm/pub), where we can see that, RecBert misinterprets the instruction "walk up the stairs" too literally and ascends any staircase it encounters, including the one in Step 4, showing a fundamental misunderstanding of the intended navigation path.
> >
> > In **Example #3**, we can see that our LangNav can generalize to the outdoor scenario, given that all of the training trajectories are sampled from the indoor scenarios. On the other hand, in the same example (as seen in [RecBert's Navigation](https://docs.google.com/document/d/e/2PACX-1vSDufWgGM-0r60U1dtxg7DfeF-uEsI_z2WIkK9UyyBp2RG2zIXpgYpAERFDtDchUphXb00_Gihzi7sm/pub)), RecBert, following identical instructions, becomes trapped in an alley, pacing back and forth. This comparison highlights LangNav's superior innate understanding that the most effective strategy in an alley is to continue straight ahead to find a way out.
> >
> > We hope the above examples could address the reviewer's concern!
> >
> > Furthermore, we strongly agree that a mechanism to correct the language-based representation failure is essential for this direction in the future. We will keep working on this. We sincerely appreciate your acknowledgment of the significance of our research!

---

### Official Review · Reviewer_2DEF · 2023-11-01

**Soundness:** 3 good
**Presentation:** 3 good
**Contribution:** 2 fair
**Rating:** 8
**Confidence:** 5

**Summary:**

This paper introduces LangNav, which explores using large language models to address vision-language navigation (VLN). LangNav applies off-the-shelf vision systems to create textual descriptions for the agent's observations, formulates prompts containing observations and decisions in navigation to query GPT-4 for synthesizing new trajectories in text form, and fine-tunes LLaMA on those trajectories to learn VLN. The paper shows that using only a few (#10/100) gold samples, GPT-4 can generate high-quality data that boosts the LLaMA performance in downstream navigation tasks. Moreover, the experiment of transferring LangNav pre-trained in synthetic ALFRED environments to photo-realistic MP3D environments demonstrates that language can be used to bridge the visual domain gap.

**Strengths:**

- This paper studies an emerging and valuable problem of applying LLMs to language-guided visual navigation to enjoy LLMs' generalization power and bridge the visual domain gap. The research is well-motivated.
- The proposed 2-phrase synthetic trajectory generation method is novel and very efficient.
- As a researcher in this topic, I am very impressed and excited to see that small LMs like LLaMA-7B can be tuned to solve VLN tasks, which is, in fact, very difficult.
    - Without complicated prompt engineering and with an effective discretization of the agent's observations (action options), LLaMA2-7B is tuned to outperform NavGPT on R2R-VLN, which the latter uses the very large and powerful GPT-4.
    - The modification on Imitation Learning helps the policy to learn to recover from deviations from the optimal path, demonstrating the potential flexibility in supervising LMs to adapt downstream tasks.
- Experiments on transferring LangNav pre-trained in ALFRED environments to MP3D environments justify the argument of using language to bridge the visual domain gap, suggesting the potential of building a generalizable agent to address visual navigation in distinct environments using different forms of instructions.

**Weaknesses:**

- The technical focus of this paper, including synthesizing trajectories in text form using GPT-4 with a few "gold samples" and tuning LLaMA for text-based navigation, is somewhat misaligned with the core argument of "using language as perceptual representation".
    - Essentially, "using language as perceptual representation" introduced in this paper is nothing but using image captions and object text labels to replace visual inputs, which is a very common practice in many V&L research and particularly the same as the important baseline NavGPT (Zhou et al., 2023).
    - The success in synthesizing trajectories with GPT-4 and tuning LLaMA should be attributed more to the very powerful pre-trained large models themselves. This success does not justify the benefit of using language but is more like a compromise of LLMs only take textual inputs. Moreover, I might be able to create an R2R textgame without any actual images (like Jericho [1]) to sample trajectories.

- I do agree that using language is promising in bridging domain gaps (e.g., different forms of instructions, 2D-3D semantics, and the Sim-Real visual gap presented in this paper), but it is unclear how far this method (of using language alone) can go.
    - As mentioned in this paper, the proposed method is limited by the off-the-shelf visual captioning and detection models. The captions could be inaccurate, not comprehensive, or ambiguous (e.g., the eye chart vs. picture example). And the ability of LLMs to reason very detailed descriptions is unclear.
    - There is an emerging trend of developing visual-LLMs (VLMs) that should be able to preserve visual clues better, which potentially could reduce the value of LangNav in future research and application. It is not very clear to me how language-only for VLN can be further extended.
    - As aware by the authors, there are many recent works on automatically synthesizing 3D environments that can be used for embodied AI training (e.g., Text2Room [2], Pathdreamer [3,4], ProcTHOR [5], etc.), and VLN data generation pipeline (e.g., MARVAL [6], AutoHM3D [7], ScaleVLN [8], etc.). The cost of building and annotating these embodied data has been significantly reduced. In ScaleVLN, the fine-tuned agent achieves 81% SR compared to LangNav 45.6% SR. Hence, I do think the performance potential of the proposed LangNav needs more justification.

- The experiments comparing LangNav to RecBERT and DUET are not strictly fair.
    - LangNav not only uses the 10/100 gold samples in fine-tuning but also the 10,000 synthetic trajectories. Those 10,000 data should not be neglected. From Table 1, LLaMA2-7B (LangNav without the 10,000 data) performs very similarly to RecBERT and DUET.
    - A more rigorous way is to create the same amount of "fake" data (using some automatic methods) to train RecBERT and DUET, although I am aware that such fake visual-text data can hardly be created at the same cost, and it is hard to define "fairness" between "language-only" and "language-visual" data.

- I believe one of the most important reasons for introducing LLMs to embodied AI tasks is to leverage LLMs' exceptional language understanding, task planning, tool-using, explicit reasoning with commonsense knowledge, communication, etc., abilities.
    - However, it is unclear in this paper whether LangNav shows any of the above abilities (e.g., using commonsense to navigate).
    - It is likely that after tunning LLaMA for VLN, the model lost these abilities, and LLaMA is simply applied as a good initialization for learning the downstream R2R task.

- The claim "Sim-to-Real" for AI2THOR-ALFRED to MP3D-R2R is very misleading. R2R is still simulated, except that it uses photo-realistic images. "Sim-to-Real" usually refers to deployment to a physical robot in the real world.

[1] Interactive Fiction Games: A Colossal Adventure. Hausknecht et al., AAAI 2020.

[2] Text2room: Extracting textured 3D meshes from 2D text-to-image models. Höllein et al., ICCV 2023.

[3] Pathdreamer: A World Model for Indoor Navigation. Koh et al., ICCV 2021.

[4] Simple and Effective Synthesis of Indoor 3D Scenes. Koh et al., AAAI 2023.

[5] ProcTHOR: Large-Scale Embodied AI Using Procedural Generation. Deitke et al., NeurIPS 2022.

[6] A New Path: Scaling Vision-and-Language Navigation with Synthetic Instructions and Imitation Learning. Kamath et al., CVPR 2023.

[7] Learning from unlabeled 3D environments for vision-and-language navigation. Chen et al., ECCV 2022.

[8] Scaling Data Generation in Vision-and-Language Navigation. Wang et al., ICCV 2023.

**Questions:**

Questions without a star (*) are not essential to my evaluation of this paper, but I still hope the authors can kindly and patiently respond to them. Please also address my concerns mentioned in Weaknesses.

1. How important are the image captions and the object labels in text descriptions?
    - How do the trajectory generation and VLN results change without any of them?
    - Does history $H$ have both the image captions and object labels? Or just image captions? This is not very clear from page 3.

2. (*) There are two important conclusions that lack convincing evidence.
    - (*) I was very impressed by the example in Figure 3. I tried to draw out the described structure, and it is indeed very spatially consistent. However, there is no numerical analysis to support the three claims of the quality of all generated trajectories (Strong Prior, Spatial Consistency, and Descriptive). Is this a general observation? I wonder if the authors can provide any quantitative results.
    - (*) I am also not fully convinced by the visualizations in Figure 4, which seems like two randomly picked examples. It is hard to say whether "eye chart" and "picture" are closer in visual feature space or text space. I wonder if the authors can give more comprehensive statistical analyses on the success/failure cases of LangNav vs. RecBERT.

3. (*) Why different LMs, LLaMA2 and LLaMA1 are used for R2R and ALFRED-R2R experiments, respectively?

4. (*) Table 2. Why do the gains are marginal when going from 2,000 to 10,000 trajectories? Is it due to the lack of diversity in generated trajectories?
    - (*) Why not use more different scenes and gold instructions to generate trajectories? How might it affect the quality of generated trajectories? Will it improve the fine-tuned LangNav?
    - (*) I am still not fully convinced by this "extremely few gold samples" setting. Why is this necessary since we have much more R2R data and the data is not too costly to generate? How does this setting generalize to other practical navigation problems?

5. The paper claims that NavGPT uses ground-truth distance, so it has an advantage and is not strictly comparable. But, using distance of viewpoints is very reasonable in R2R because the task is defined in a discretized space (DUET also uses this information). Even in continuous environments, a waypoint predictor can be applied to return the exact distance of navigable positions [9,10,11].
    - What would be the results of training LangNav with distance information?

6. (*) Are the RecBERT and DUET in Table 1 and Table 3 pre-trained on R2R, e.g., with tasks such as masked language modeling and single action prediction? This is unclear in the paper, but it is very important in the discussion of the few-shot learning and visual domain gap.

Others:
- Figure 1. Use the same color for image C (orange) and "C" in the caption (pink).
- Typo in S5 Limitations. "table 2" should be "table 4".
- Table 4. A "dagger" sign is missing in the caption of re-implementation. Also, I believe RecBERT hasn't been pre-trained on the val_unseen set. What does this re-implementation exactly do?
- In-text citations have inconsistent formats (e.g., author names inside/outside brackets).
- Please add an ethical statement and reproducibility statement at the end of the paper.

[9] Waypoint models for instruction-guided navigation in continuous environments. Krantz et al., CVPR 2021.

[10] Sim-to-real transfer for vision-and-language navigation. Anderson et al., PMLR 2020.

[11] Bridging the gap between learning in discrete and continuous environments for vision-and-language navigation. Hong et al., CVPR 2022.

---

> ### Author Response · Authors · 2023-11-22
> **Response to Reviewer 2DEF (1/N)**
>
> We thank the reviewer 2DEF for appreciating our work!
>
> ***(1) Focus of our paper:*** We agree that more focus should be placed on the language-based representation itself. We note that our two case studies (i.e., efficient synthetic data generation and ALFRED-to-R2R transfer) actually are explicitly leveraging the fact that language can serve as an efficient and robust representation of one's perception. However, we agree that this aspect of our work should be emphasized, and will clarify this point more in the paper.
>
> Moreover, your suggestion of aligning our focus with the benefit of using language as representation inspired us to perform a third "case study", where we experiment to see whether language-based representation allows us to interpret and edit our policy. Concretely, we randomly pick 10 R2R trajectories where the LangNav fails to choose the correct direction, like the second example in Figure 4 in the paper. For the step where the LangNav acts wrong, we manually correct the incorrect captions using visual analysis and the given instructions. We provide three examples that demonstrate that, by merely correcting the incorrect captions, LangNav is able to change the decision from incorrect to correct. The original trajectory examples are in [the link to original trajectories](https://docs.google.com/document/d/e/2PACX-1vRHHEyRJXVe0kHd6N4yMqe_XjQvHthB7VAs4f-NYNWBOH_-zDCfOtFtXAmhCs2ZPzHG-28QMHQKn_Vw/pub) and the modified trajectory examples are in [the link to modified trajectories](https://docs.google.com/document/d/e/2PACX-1vSUhL58LvLpF47YKDPcwcgk7rIHGhsW9IFOeW-pWkMlBg9v_8LaF4He-rmioksk-y-czpyxKIucKik6/pub), where we mark the modified caption in ***bold***. We applied this procedure to 10 selected R2R trajectories and discovered that our LangNav was able to revise its decision correctly in **70%** of these trajectories by only rectifying incorrect captions. For the other 30% examples, the failure reason doesn't fall into the vision captions in the current step. In Example #3, the agent decides to continue moving forward in Step 2 since it does not recognize from its historical data that it has already ascended the stairs. We thank the reviewer for suggesting these experiments. We will include these results in our next version of the paper.
>
> **(2) How far can this method (of using language alone) go?** In the response to (1), we show that we can manipulate the vision captions in the prompt and discover that inaccurate visual perceptions contribute significantly to the failure cases, which indicates that although the current VLM system is not yet perfect and the performance of our LangNav is influenced by the inaccurate, not comprehensive, and ambiguous captions, we remain optimistic that future improvements in vision and text models will greatly enhance LangNav's long-term performance, especially with the great potential of VLM we see from the GPT-4V.

---

> ### Author Response · Authors · 2023-11-22
> **Response to Reviewer 2DEF (2/N)**
>
> **(3) Emerging trend of developing visual-LLMs (VLMs):** We agree that visual-LLMs could preserve more visual details compared to the language-only representations. However, using language-based representation is not contradictory to the continuous visual features. To demonstrate how the language can help *on top of* continuous visual features, we performed further experiments where we extend the RecBert by concatenating language features to the visual features to represent the candidate image. More specifically, the original RecBert uses ResNet-152 to extract the visual feature to represent the image, while our extension additionally concatenates the language feature of the caption to be the image representation. The captions are the same as what are used in LangNav and the features are extracted by BERT-base. We pre-train and fine-tune the RecBert and its extension with 100-shot training trajectories and full R2R train set. The results are listed in the tables. We can see that the language features improve the performance in both few-shot and full learning cases, which indicates that language-based features can help the visual perception module for the VLN agent.
>
> | eval env   | train trajectories | perceptual features | SR↑  | SPL↑  |
> |------------|:--------------------:|:----------------------:|------|------|
> |  val_seen  |          100         |           vision   only         | 19.9 | 18.6 |
> | val_unseen |          100         |           vision  only         | 19.0 | 17.4 |
> |  val_seen  |          100         |           vision + language       |**22.8**|**21.5**|
> | val_unseen |          100         |           vision + language        |**19.3**|**18.0**|
>
> | eval env   | train trajectories  | perceptual features | SR↑   | SPL↑  |
> |------------|:--------------------:|:----------------------:|------|------|
> |  val_seen  |      full train      |            vision  only           | 58.5 | 55.4 |
> | val_unseen |      full train      |            vision  only           | 47.1 | 43.4 |
> |  val_seen  |      full train      |           vision + language        |**61.4**|**57.1**|
> | val_unseen |      full train      |           vision + language        |**48.8**|**44.1**|
>
> These results indicate that language as a perceptual representation can provide additional benefits on top of continuous visual features. We thank the reviewer for suggesting this experiment and will include it in the next version of the paper.
>
> **(4) Fair comparison to RecBert and justification for the performance potential of LangNav:** To fairly compare the performance of our LangNav with RecBert and justify the performance potential of our LangNav, we present the results of evaluating both LangNav and RecBert which are trained on full R2R train split. We pre-train and fine-tune the RecBert only on R2R train split without any synthetic data (neither PREVALENT data nor GPT-4 synthetic trajectories). The results are listed in the table. It is observed that LangNav still lags behind RecBert in performance. Nonetheless, we attribute this gap primarily to the mismatch between visual and textual data. As highlighted in the response to point (1), the role of unclear and imprecise captions in causing failures is considerable. Therefore, we maintain a positive outlook on LangNav's future capabilities, anticipating that advancements in vision and text processing technologies will significantly boost LangNav's efficacy in the long run.
> | eval env   | train trajectories  | Method | SR↑   | SPL↑  |
> |------------|:--------------------:|:----------------------:|------|------|
> |  val_unseen |      full train     |          RecBert       | 47.1 | 43.4 |
> | val_unseen |      full train      |          LangNav       | 43.2 | 37.9 |
>
>
> **(5) Commonsense for navigation:** We agree that the capabilities of LLMs are the primary reason why we wish to leverage them and their language-based representations. To illustrate the commonsense reasoning ability of the LLM used in LangNav, we have visualized the decision-making process in a specific [navigation example](https://docs.google.com/document/d/e/2PACX-1vSDRyA2uxaSMBdQ85fCxzJfwMLAMXt-X_aJvOw_TuANWg7JF_xpOSqs1fHNJFX-Iw9uvRoTUG9Q2uc0/pub) from LangNav. For each step, we display an image showing the direction the agent faces. From Step 2 to Step 3, the agent demonstrates commonsense reasoning. Upon receiving the instruction, 'turn left, walk forward, and then turn left,' the agent interprets and integrates these directions, ultimately making a decision equivalent to 'turning left back,' as demonstrated in the example.

---

> > ### Author Response · Authors · 2023-11-22
> > **Response to Reviewer 2DEF (3/N)**
> >
> > **(6) "Sim-to-real" claim:** We thank the reviewer for pointing this out! We agree that our experiments are still conducted within a simulated environment. We will change the name to 'Sim-to-Sim' in our subsequent version.
> >
> > **Q1 How important are the image captions and the object labels in text descriptions?**
> > To investigate the importance of the object labels, we present the comparison of the LangNav trained on the full R2R train set w/ and w/o object labels. The results don't involve the random approach which is described in section 3.3. The boost in performance by adding object labels is marginal and using captions alone is more efficient considering the prompt length. Therefore, we use only image captions in the case study 1 and table 4. The history ***H*** in case study 1 has only image captions. In case study 2, ***H*** has both the image captions and object labels.
> >
> > | eval env   | train trajectories  | observations | SR↑   | SPL↑  |
> > |------------|:--------------------:|:----------------------:|------|------|
> > | val_unseen |      full train     |captions| 35.3 | 33.5 |
> > | val_unseen |      full train      |captions+object labels| 36.4 | 34.6 |
> >
> > **(*) Q2 Is there any quantitative result for the quality of generated trajectories?**
> > Yes! To compare the quality of the generated trajectories, we conduct an experiment where we prepare 93 training trajectories for both real-world data and synthetic data, and finetune the LLaMA2-7B model. In order to control the scene constraint, the 93 real-world trajectories all originate from the same scan used to sample the 10 trajectories for prompting LLMs. The evaluation results are shown in the table. The evaluation shows that, under the same constraints of the training scene diversity, synthetic trajectories have comparable or even better quality with the real-world trajectories. We infer the gain of the synthetic data is from the more accurate language description of the scene and the higher abundance of the object concepts due to GPT-4's hallucination.
> >
> > |  Eval Env  | Training data |   NE↓  |   SR↑  |  SPL↑  |
> > |:----------:|:-------------:|:-----:|:-----:|:-----:|
> > |  val_seen  |   real-world  | 10.40 | 14.89 | 10.60 |
> > |  val_seen  |    sythetic   | **8.81**  | **16.85** | **13.54** |
> > | val_unseen |   real-world  | 10.16 | 14.09 |  9.54 |
> > | val_unseen |    sythetic   | **8.29**  | **17.41** | **14.59** |
> >
> > **(*) Q3 Can the authors give more comprehensive statistical analyses on the success/failure cases of LangNav vs. RecBERT?**
> > Yes! As we present in the response to (1), we randomly pick 10 R2R trajectories in which RecBert succeeds while our LangNav fails. We focus on the first step where the LangNav chooses the incorrect direction, like the second example in Figure 4. For the step where the LangNav acts wrong, we manually correct the incorrect captions using visual analysis and the given instructions. Examples can be found in [the link to original trajectories](https://docs.google.com/document/d/e/2PACX-1vRHHEyRJXVe0kHd6N4yMqe_XjQvHthB7VAs4f-NYNWBOH_-zDCfOtFtXAmhCs2ZPzHG-28QMHQKn_Vw/pub) and [the link to modified trajectories](https://docs.google.com/document/d/e/2PACX-1vSUhL58LvLpF47YKDPcwcgk7rIHGhsW9IFOeW-pWkMlBg9v_8LaF4He-rmioksk-y-czpyxKIucKik6/pub). We applied this procedure to 10 selected R2R trajectories and discovered that our LangNav was able to revise its decision correctly in 70% of these trajectories. For the other 30% examples, the failure reason doesn't fall into the vision captions in the current step. For example in Example #3, the agent decides to continue moving forward in Step 2 since it does not recognize from its historical data that it has already ascended the stairs.
> >
> > **(*) Q4 Why different LMs, LLaMA2 and LLaMA1 are used for R2R and ALFRED-R2R experiments, respectively?**
> > This is because of the timeline when the experiments were conducted. The experiments of ALFRED-R2R and full R2R were conducted before LLaMA2 was released. We will update both the results with LLaMA2 in our next version of the paper.

---

> > > ### Author Response · Authors · 2023-11-22
> > > **Response to Reviewer 2DEF (4/N)**
> > >
> > > **(*) Q5 Why do the gains are marginal when going from 2,000 to 10,000 trajectories? Is it due to the lack of diversity in generated trajectories?**
> > > Thanks for pointing this out. We infer that the fast convergence of synthetic data is due to the use of only 10 real trajectories from a single scene to prompt LLMs. For example, here are four generated instructions from the same output of the LLM:
> > >
> > > > *1. Start from the main entrance door, pass the living room, and enter the kitchen on your right. Locate the refrigerator, then turn left and stop just before the dining table.
> > > > 2. Navigate from the couch in the living room, move towards the mantel, and then stop next to the fireplace. Avoid any furniture and obstacles on your path.
> > > > 3. Begin at the foot of the bed in the master bedroom. Walk forward and enter the attached bathroom. Once you're inside, stop next to the bathtub.
> > > > 4. Start in the family room, walk towards the TV, then turn right and pass the bookshelf. Stop when you reach the large bay window overlooking the garden.*
> > >
> > > We can see from the above synthetic instructions that (a) patterns of the synthetic instructions are similar, which are like "Start from place A, go pass place B, stop at place C", (b) scenes are limited to the living area and a single floor, however, the R2R tasks always require the agent navigating across floors and in some non-living area.
> > >
> > > This indicates that the underperformance of the generated trajectories compared to real trajectories is due to scene diversity. To further investigate the influence of the scene diversity, we conduct an experiment where we use 1000 navigation instructions sampled from various R2R scenes to prompt GPT-4-turbo to generate 2000 synthetic trajectories. The scaling results are listed in the table. We can see that although the 2000 trajectories generated by GPT-4-turbo are not of the same quality as those generated by GPT-4, scaling up using these trajectories outperforms the results from the 10000-trajectory set.
> > >
> > > | Number of synthetic trajectories | Seed trajectories |         LLM         |  OSR↑ | SR↑   | SPL↑  |
> > > |--------------------------------|-----------------|-------------------|----|------|------|
> > > |               2,000              |         10        | GPT-4               | 42.2 | 31.1 | 26.6 |
> > > |               2,000              |         1,000        | GPT-4-turbo               | 42.9 | 24.9 | 19.6 |
> > > |           2,000 + 2,000          |     10 + 1,000    | GPT-4 + GPT-4-turbo | **43.2** | **32.6** | **28.3** |
> > > |              10,000              |         10        | GPT-4               | 41.9 | 31.6 | 27.5 |
> > >
> > > **(*) Q6 How does the low-data setting generalize to other practical navigation problems?**
> > > Another practical navigation problem to motivate the setting is navigation in some exotic environments (e.g. office environments, supermarkets, or industrial environments) as suggested by the reviewer PNyi, where the training data is scarced. We show how we generate synthetic data in exotic environments with the practice below.
> > >
> > > We handcraft a trajectory in a real office environment and then prompt GPT-4 to generate synthetic trajectories within the scope of office environment by following the pipeline in Figure 2. Here are the sampled real trajectory and two generated synthetic trajectories: [link to real/synthetic trajectories in office environments](https://docs.google.com/document/d/e/2PACX-1vRnsNCGDadAKsfYvpUD-q9dXGd8qGGkkJ04901S8ocE_52WE5ns_9C1KD_VX1LlznhcSmKqot3VNfxj/pub).
> > >
> > > We can see that the synthetic trajectories contain abundant common object-scene correlations in office environments (example), exhibit great spatial consistency (example), and incorporate a significant amount of captions and objects that do not directly relate to the given instruction. We nonetheless think this is an interesting experiment and will update the paper with these qualitative results on exotic environments.
> > >
> > > **Q7 What would be the results of training LangNav with distance information?**
> > > In our experiments, we have observed that the object labels are sometimes noisy because of wrong detections and irrelevant objects. Also including object labels largely increases the length of the prompt which makes the training and inference not efficient. Thus, we believe the distance information would definitely help the performance of LangNav from the aspect of filtering the distant objects to making the representation more informative and efficient.

---

> > > > ### Author Response · Authors · 2023-11-22
> > > > **Response to Reviewer 2DEF (5/N)**
> > > >
> > > > **(*) Q8 Are the RecBERT and DUET in Table 1 and Table 3 pre-trained on R2R?**
> > > > That's a great question! The RecBERT baselines in Table 1 and Table 3 is pre-train on 10/100 trajectories from R2R with masked language modeling (MLM) and single action prediction (SAP) tasks. The DUET baselines in Table 1 is pre-trained on 10/100 trajectories with MLM, SAP, and masked region classification (MRC) tasks. We will add the clarifications in our next version of the paper.
> > > >
> > > > **Q9  What does the re-implementation of RecBERT exactly do?**
> > > > As the RecBERT uses PREVALENT [12] pre-trained weights to initialize the model, we assume it has been pre-trained on the val_unseen set. In the PREVALENT paper, the authors say that the pre-trained dataset includes 104K image-text-action triplets from R2R in section 4.4, which equals the number of total triplets in "shortest_1.json", "shortest_2.json", and "shortest_3.json" (https://drive.google.com/drive/folders/1tvg8Kuu5Q1wfFGIa-ha8NNqv0Nd6x-EO). However, "shortest_2.json" and "shortest_3.json" are extracted from val_seen set and val_unseen set. We also verified that with the authors of PREVALENT [12]. The re-implementation removes the val_seen and the val_unseen data from the pre-training.
> > > >
> > > > **Others:**
> > > > Thanks to the reviewer for pointing these out! We will revise the color issue in Figure 1, revise the typo in section 5, add the "dagger" sign to the caption, fix the citation formats, and add the ethical statement and reproducibility statement in our next version of the paper!
> > > >
> > > > [12] Hao W, Li C, Li X, et al. Towards learning a generic agent for vision-and-language navigation via pre-training.

---

> ### Comment · Reviewer_2DEF · 2023-11-22
> **Final Rating**
>
> I want to thank the authors for the extremely detailed response to my comments. I really appreciate the great efforts in delivering additional experiments. Many of my questions have been nicely addressed. After reading all the reviews and responses, I decided to keep my rating as Accept. Meanwhile, I hope the authors can experiment and discuss more my major concern of
>
> ```
> I believe one of the most important reasons for introducing LLMs to embodied AI tasks is to leverage LLMs' exceptional language understanding, task planning, tool-using, explicit reasoning with commonsense knowledge, communication, etc., abilities.
>
> However, it is unclear in this paper whether LangNav shows any of the above abilities (e.g., using commonsense to navigate).
> It is likely that after tunning LLaMA for VLN, the model lost these abilities, and LLaMA is simply applied as a good initialization for learning the downstream R2R task.
> ```

---

> ### Author Response · Authors · 2023-11-22
> **More examples of LangNav's ability**
>
> We are glad that our rebuttal has addressed most of the reviewer's concerns! To show more about the ability of LangNav, we show three more examples attached in [LangNav's Navigation](https://docs.google.com/document/d/e/2PACX-1vTVmWvp1gHwLcc8ZY8ybMZ-6uDannhGZ4pBn2GO2lxwiE8fW_zTcJi-mPGxaWzz3BP1MB0vvEu0T-_T/pub).
>
> In **Example #1**, we have observed that, although the instruction of where to stop is vague and ambiguous ("Wait at the entrance."), our LangNav agent manages to stop in front of the double wooden doors which is the most likely to be the entrance.
>
> In **Example #2**, from Step 2 to Step 4, our LangNav agent erroneously enters a bathroom on the right in Step 2, but then corrects its course by leaving the bathroom and proceeding to the living room. In contrast, we have also visualized the trajectory (Example #2) of RecBert under the same instructions in [RecBert's Navigation](https://docs.google.com/document/d/e/2PACX-1vSDufWgGM-0r60U1dtxg7DfeF-uEsI_z2WIkK9UyyBp2RG2zIXpgYpAERFDtDchUphXb00_Gihzi7sm/pub), where we can see that, RecBert misinterprets the instruction "walk up the stairs" too literally and ascends any staircase it encounters, including the one in Step 4, showing a fundamental misunderstanding of the intended navigation path.
>
> In **Example #3**, we can see that our LangNav can generalize to the outdoor scenario, given that all of the training trajectories are sampled from the indoor scenarios. On the other hand, in the same example (as seen in [RecBert's Navigation](https://docs.google.com/document/d/e/2PACX-1vSDufWgGM-0r60U1dtxg7DfeF-uEsI_z2WIkK9UyyBp2RG2zIXpgYpAERFDtDchUphXb00_Gihzi7sm/pub)), RecBert, following identical instructions, becomes trapped in an alley, pacing back and forth. This comparison highlights LangNav's superior innate understanding that the most effective strategy in an alley is to continue straight ahead to find a way out.
>
> We hope the above examples could address the reviewer's concern!

---

### Official Review · Reviewer_z42V · 2023-11-03

**Soundness:** 2 fair
**Presentation:** 2 fair
**Contribution:** 2 fair
**Rating:** 3
**Confidence:** 4

**Summary:**

The paper proposes using language as the perceptual representation of the navigation agent environments in the domain of vision-and-language navigation. Such an approach would allow using LMs (language models) to model the navigation task as a pure language task. The visual scenes are expressed as text descriptions from vision models (such as a captioning model). Two empirical studies are shown -- first shows efficient synthetic data generation using just language by using prompt engineering on a LLMs (such as GPT-4), second shows that language enables more efficient sim-to-real transfer compared to vision models.

**Strengths:**

* The Case Study 2 is very interesting: it shows that when transferring from a simulated environment (AI2THOR) to real environment (R2R), using a model that relies on vision input generalizes worse than a model that relies solely on text input. This is also intuitive in that while actual pixel-level details can differ, the environments at semantic level should have similarities and the sim-to-real transfer should work better with a modality that learns semantic representation of the environment.

**Weaknesses:**

* The Case Study 1 is not very informative. The LangNav not only benefits from synthetic instructions but also benefits from the number of samples used for finetuning. While baselines are finetuned on only 10/100 real-world trajs, the LangNav model is finetuned on 10/100 real-world + 10,000 synthetic trajs. For a fair comparison and to truly understand the quality of synthetic instructions, we need a study something like:

  * A: trained on 10k real-world trajs
  * B: trained on 10k synthetic trajs
  * C: trained on 5k (sampled) real-world + 5k (sampled) synthetic trajs

  In the above setup, comparing A vs B will indicate the quality of synthetic trajs. Comparing A vs C will indicate if synthetic trajs can
  complement a smaller number of real-world trajs to achieve the same quality.

* Given that no new models or methods are developed, the key contributions of this paper are relatively light.

Overall, I think this is a good workshop paper to motivate future work in this direction, but in itself, does not meet the bar as a conference paper.

**Questions:**

Some of the points below are more like suggestions for future improvement, rather than immediate action items:

* In 4.2, it is worth reiterating the differences between RecBERT and LangNav models.

* It would be more interesting if there can be a more end-to-end learning setup with vision-to-text module part of the training process.

* In 3.3, a strategy is described to make the model robust to deviations from logged (imitation) policy. A simple and more principled strategy would use something similar to RLHF -- use trained model policy pi_{t} to infer on trajectories from training set, assign rewards (simple goal based as an example) for the inferred trajectories, keep the ones with high reward (some threshold) and train pi_{t+1} on union of existing training dataset and those with high reward.

---

> ### Author Response · Authors · 2023-11-19
> **Response to Reviewer z42V (1/N)**
>
> We would like to thank Reviewer z42V for these valuable comments.
>
> ***(1) Comparison beween synthetic data and realistic data:*** To address the reviewer's corcern about the quality of the synthetic data, we conduct an experiment where we prepare 93 training trajectories for both real-world data and synthetic data, and finetune the LLaMA2-7B model. In order to control the scene constraint, the 93 real-world trajectories all originate from the same scan used to sample the 10 trajectories for prompting LLMs. The evaluation results are shown in the table. The evaluation shows that, under the same constraints of the training scene diversity, synthetic trajectories have comparable or even better quality with the real-world trajectories. We infer the gain of the synthetic data is from the more accurate language description of the scene and the higher abundance of the object concepts due to GPT-4's hallucination.
>
> |  Eval Env  | Training data |   NE↓  |   SR↑  |  SPL↑  |
> |:----------:|:-------------:|:-----:|:-----:|:-----:|
> |  val_seen  |   real-world  | 10.40 | 14.89 | 10.60 |
> |  val_seen  |    sythetic   | **8.81**  | **16.85** | **13.54** |
> | val_unseen |   real-world  | 10.16 | 14.09 |  9.54 |
> | val_unseen |    sythetic   | **8.29**  | **17.41** | **14.59** |
>
> ***(2) Contributions of the paper:***  We respectfully disagree with the reviewer's assessment that our paper makes a relatively light contribution due to the lack of development of new models or methods. We would like to remind the reviewer that our core contribution is not a model or method (since, as the reviewer notes, we make use of existing methods/models); rather, our core contribution is to suggest that language itself can serve as a perceptual representation for navigation especially in low-data regimes, which has been underexplored compared to uses of language in other embodied tasks. We conduct two experiments that exploit the use of language via (1) synthetic data generation and (2) sim-to-real transfer to improve VLN.
>
> Moreover,  we conducted additional experiments where we augment a traditional vision-based navigation model with language features. This is a "new model" insofar as existing VLN systems only rely on visual perception features. More specifically, the original RecBert uses ResNet-152 to extract the visual feature to represent the image, while our extension additionally concatenates the language feature of the caption to be the image representation. The captions are the same as what are used in LangNav and the features are extracted by BERT-base. We pre-train and fine-tune the RecBert and its extension with 100-shot training trajectories and full R2R train set. The results are listed in the tables below.
>
>  | eval env   | train trajectories | perceptual features | SR↑   | SPL↑  |
> |------------|:--------------------:|:----------------------:|------|------|
> |  val_seen  |          100         |           vision   only         | 19.9 | 18.6 |
> | val_unseen |          100         |           vision  only         | 19.0 | 17.4 |
> |  val_seen  |          100         |           vision + language       |**22.8**|**21.5**|
> | val_unseen |          100         |           vision + language        |**19.3**|**18.0**|
>
> | eval env   | train trajectories  | perceptual features | SR↑   | SPL↑  |
> |------------|:--------------------:|:----------------------:|------|------|
> |  val_seen  |      full train      |            vision  only           | 58.5 | 55.4 |
> | val_unseen |      full train      |            vision  only           | 47.1 | 43.4 |
> |  val_seen  |      full train      |           vision + language        |**61.4**|**57.1**|
> | val_unseen |      full train      |           vision + language        |**48.8**|**44.1**|
>
> We can see that the language features improve the performance in both few-shot and full learning cases, which indicates that language-based features can help the visual perception module for the VLN agent. We hope that this addresses the reviewer's concerns about the modeling contribution of the work.
>
> ***(3) Reiterating the differences between RecBERT and LangNav models:*** We thank the reviewer for the suggestion. We will reiterate the detailed differences between RecBERT and LangNav models in the revised version of the paper, ensuring that the distinctions are clearly outlined.

---

> ### Author Response · Authors · 2023-11-19
> **Response to Reviewer z42V (2/N)**
>
> ***(4) End-to-end learning setup with vision-to-text module:*** We think that end-to-end learning for the vision-to-text module would be an interesting direction to explore! With these settings, the vision-to-text module can benefit from LM policy and learn to focus more on navigation-relevant details.  To investigate how a better vision-to-text module can help improve the VLN performance, and interpret the failure reason, we randomly pick 10 R2R trajectories where the LangNav fails to choose the correct direction, as illustrated by the second example in Figure 4 of the paper. For the step where the LangNav acts wrong, we manually correct the incorrect captions using visual analysis and the given instructions. We provide three examples which demonstrate that, by merely correcting the incorrect captions, LangNav is able to change the decision from incorrect to correct. The original trajectory examples are in [the link to original trajectories](https://docs.google.com/document/d/e/2PACX-1vRHHEyRJXVe0kHd6N4yMqe_XjQvHthB7VAs4f-NYNWBOH_-zDCfOtFtXAmhCs2ZPzHG-28QMHQKn_Vw/pub) and the modified trajectory examples are in [the link to modified trajectories](https://docs.google.com/document/d/e/2PACX-1vSUhL58LvLpF47YKDPcwcgk7rIHGhsW9IFOeW-pWkMlBg9v_8LaF4He-rmioksk-y-czpyxKIucKik6/pub), where we mark the modified caption in ***bold***. We applied this procedure to 10 selected R2R trajectories and discovered that our LangNav was able to revise its decision correctly in **70%** of these trajectories by only rectifying incorrect captions. The results indicate that a navigation-oriented vision-to-text module is essential for VLN tasks. End-to-end learning is a promising way to learn such a model.
>
> ***(5) Methods like RLHF to make the model robust to deviation:*** Alongside the strategy outlined in Section 3.3, we conducted an external experiment using the DAgger algorithm for imitation learning. This process involves utilizing a trained language model (LM) policy to infer trajectories within the training environment. These inferred trajectories are then combined with the existing training dataset to form a new training set. The foundational concepts of the DAgger algorithm and the reviewer's proposed strategy bear a strong resemblance. In our comparison table, we illustrate the performance differences among the baseline LM policy, the LM policy enhanced through DAgger, and our method designed to address deviations. The findings demonstrate that although DAgger substantially improves VLN performance, the approach we developed, as described in Section 3.3, ultimately yields superior results over the DAgger algorithm.
>
> | eval env   |     policy     |  OSR↑ |  SR↑  |  SPL↑ |
> |------------|:--------------:|:----:|:----:|:----:|
> | val_unseen | base LM policy | 43.3 | 38.6 | 36.9 |
> | val_unseen |     DAgger     | 47.2 | 40.7 | 37.8 |
> | val_unseen |      Ours      | **50.3** | **43.2** | **37.9** |

---

> ### Author Response · Authors · 2023-11-22
>
> Dear Reviewer z42V,
>
> Thank you for your constructive comments and suggestions. As the discussion phase is nearing its end, we wondered if you might still have any concerns that we could address. We believe our response addressed all your questions/concerns, and hope that the work's impact and results are better highlighted with our responses. Thank you!
>
> Best wishes,
>
> Authors

---

> ### Comment · Reviewer_z42V · 2023-12-04
> **[Final Reviewer Ratings]**
>
> The detailed responses provided by the authors to all the reviews is greatly appreciated. After going through the author responses as well as other reviewers feedback, there are several outstanding concerns with the main hypotheses of the paper. While the additional experiments conducted by the authors are appreciated, they do not change the position of the paper which has been correctly called into question in other reviews. In response to the first weakness, authors conducted additional experiment using 93 training trajectories. While the author efforts are commendable, it is not clear how much conclusions we can draw from such a small scale experiment. Overall, I am hesitant to change my rating from "3: reject, not good enough".

---

### Official Review · Reviewer_Sthf · 2023-11-05

**Soundness:** 2 fair
**Presentation:** 2 fair
**Contribution:** 2 fair
**Rating:** 5
**Confidence:** 3

**Summary:**

The paper introduces LangNav, a novel approach to vision-based robotic navigation guided by natural language descriptions. LangNav uses off-the-shelf visual models to convert it into language descriptions and leverages language models to make navigation decisions. The proposed approach formulates the robotic navigation problem to leverage the capabilities of a language model for decision making. Actions are formulated to be selected from a finite set of “observations”. The paper proposes efficient synthetic dataset generation and transfer from ALFRED to R2R environments that have different 3D visual appearances.
While LangNav shows promise in improving navigation performance, further research and empirical experiments are needed to address these concerns and validate its practicality in complex and diverse robotic environments.

**Strengths:**

1. The paper demonstrates how few-shot examples can be used to generate synthetic data and train a language-based navigation policy, thereby highlighting “language itself as a perceptual representation space”.
1. The authors demonstrate that a small LM like Llama-7B  outperforms a vision-based agent that is finetuned on the same seed data.
1. The paper demonstrates “language as a domain-invariant representation” by training agent on ALFRED and transferring it to R2R environments.

**Weaknesses:**

1. Incomplete Visual/Spatial Information: The image captioning and object detection modules may not provide all the necessary visual and spatial information needed to successfully complete the navigation task. Since the dataset contains trajectories that were constructed based on the  vision systems to obtain the captions/objects, such a scenario will never be encountered in the training/test set. But this raises concerns about the reliance of LangNav for accurate and relevant visual captioning/detection in real-world scenarios.
1. Applicability to Real-World Robots: Just like ImageNet, which organizes images according to the WordNet hierarchy but does not necessarily contain a wide range of real-world objects, it seems the scope of task and navigation instructions that were used to train are those where the captioning model provides task relevant descriptions. This raises concerns about the applicability of the LangNav approach to real-world robotic systems, where the environment and tasks can be much more complex and varied than what the training data might represent.
1. Simulation-to-Simulation Transfer: It seems inappropriate to label the R2R environment, based on Matterport scans, as "real-world." There is a lack of empirical experiments or evidence to support the effectiveness of transferring policies learned in simulated environments to genuinely real-world scenarios. This raises questions about the robustness and practicality of the proposed approach in real-world robotics applications.

**Questions:**

1. Episode Length and Performance: Are there empirical experiments or findings related to the performance of the LangNav system concerning the length of the navigation episode? Does the model's performance vary with longer or shorter episodes, and if so, how?
1. Expressivity of Observations and Actions: The expressivity of observations and actions seems to influence the size and structure of the prompt, so what role does it play in the overall task performance?
1. Prompt History and Reasoning: Can you provide more details on how updating the history in the prompt affects the reasoning in the language model? Are there specific mechanisms or techniques used for incorporating historical information, and how does this impact the model's performance?
1. Biases in Synthetic Data Generation: Given that "LangNav underperforms baselines in data-rich regimes," can you elaborate on any potential biases in the synthetic data generation process? Are there challenges or limitations in scaling up the generation of synthetic data for training the LangNav system, and how might these biases be addressed?
1. Minor clarifications/suggestions:
- real-world environment (R2R)? Is it really real-world? → a more apt claim could be simulators with perceptually different representations or simulation to simulation transfer.


- In Section 4.1.1, “a real demonstration instruction and trajectory” is unclear. Does this mean rolling out the episode in the chosen simulated environment?


- In Section 7, “real-word” -> “real-world”

---

> ### Author Response · Authors · 2023-11-19
> **Response to Reviewer Sthf (1/N)**
>
> We would like to thank Reviewer Sthf for these valuable comments.
>
> ***(1) Incomplete visual/spatial information:*** This is an interesting point, and we agree with the reviewer's concern regarding the influence of incomplete visual/spatial information in the real world. However, the fact that we obtain nontrivial performance when we transfer a policy trained on ALFRED to R2R (even with zero R2R examples) indicates that the LangNav agents do not overfit to the captions/objects from the domain, and are able to generalize the navigation knowledge across domains even in the presence of incomplete visual/spatial information. Moreover, we show below (in our response to (2)) that even in the presence of continuous visual features, language captions can still improve performance.
>
> ***(2) Applicability to real-world robots:*** We acknowledge that real-world scenarios are much more complex than the R2R environment. Beyond the more complex and varied scene configurations, objects in the real world are also dynamic, rather than static. Because of this, we strengthen our belief that language is an important perceptual representation to bridge the gap (a view that both Reviewer 2DEF and Reviewer PNYi agree with). Our Case Study 2 also demonstrates that combining the high-level abstraction characteristic of language with the common-sense reasoning ability of LLMs is a more promising way to transfer knowledge between different environments.
>
>
> On the other hand, we also find that language-based features can also help the traditional visual perception module of the navigation agent. To further explore how the language can help *on top of* continuous visual features, we performed further experiments where we extend the RecBert [1] by concatenating language features to the visual features to represent the candidate image. More specifically, the original RecBert uses ResNet-152 to extract the visual feature to represent the image, while our extension additionally concatenates the language feature of the caption to be the image representation. The captions are the same as what are used in LangNav and the features are extracted by BERT-base [10]. We pre-train and fine-tune the RecBert and its extension with 100-shot training trajectories and full R2R train set. The results are listed in the tables. We can see that the language features improve the performance in both few-shot and full training set cases, which indicates that language-based features can help the visual perception module for the VLN agent.
>
> | eval env   | train trajectories | perceptual features | SR↑   | SPL↑  |
> |------------|:--------------------:|:----------------------:|------|------|
> |  val_seen  |          100         |           vision   only         | 19.9 | 18.6 |
> | val_unseen |          100         |           vision  only         | 19.0 | 17.4 |
> |  val_seen  |          100         |           vision + language       |**22.8**|**21.5**|
> | val_unseen |          100         |           vision + language        |**19.3**|**18.0**|
>
> | eval env   | train trajectories  | perceptual features | SR↑   | SPL↑ |
> |------------|:--------------------:|:----------------------:|------|------|
> |  val_seen  |      full train      |            vision  only           | 58.5 | 55.4 |
> | val_unseen |      full train      |            vision  only           | 47.1 | 43.4 |
> |  val_seen  |      full train      |           vision + language        |**61.4**|**57.1**|
> | val_unseen |      full train      |           vision + language        |**48.8**|**44.1**|
>
> These results indicate that language as a perceptual representation can provide additional benefits on top of continuous visual features. We will include these results in the paper.
>
>
>
> ***(3) Sim2sim transfer:*** We thank the reviewer for pointing this out! We agree that our experiments are still conducted within a simulated environment. We will change the name to 'Sim-to-Sim' (or 'domain transfer') in our subsequent version.
>
> ***Q1 Does the model's performance vary with longer or shorter episodes, and if so, how?***
>
> Yes, the model's performance varies with the length of the episode. We evaluated the LangNav model, which was fine-tuned on 2,000 GPT-4 synthetic trajectories and 100 real-world trajectories, to calculate the success rate (SR) and navigation error (NE). The results are listed in the table. As can be seen from the table, as the episode length increases, the task becomes more complex, and the navigation performance decreases, which aligns with our intuition.
>
> | Trajectory Steps |   4   |   5   |   6   | 7     | Overall |
> |:----------------:|:-----:|:-----:|:-----:|-------|---------|
> |        NE↓        | 3.56  | 5.98  | 6.88  | 8.25  | 6.99    |
> |        SR↑        | 54.2 | 37.1 | 35.7 | 28.3 | 33.8   |

---

> > ### Author Response · Authors · 2023-11-19
> > **Response to Reviewer Sthf (2/N)**
> >
> > ***Q3 How updating the history in the prompt affects the reasoning in the language model? Are there specific mechanisms or techniques used for incorporating historical information, and how does this impact the model's performance?***
> >
> > The method we use to update the history is straightforward: we simply expand the prompt with the output action at the current step, without any additional complexities. The key aspect is to retain all the history information from the beginning within the prompt. Here, we present experimental results comparing the single-step action accuracy between two different models: (1) the LLaMA-7B model fine-tuned on prompts with full history, and (2) the LLaMA-7B model fine-tuned on prompts with a maximum of 3 history steps. The evaluation was conducted on the R2R val unseen split. As can be seen from the table, having full history information is essential for the model's accuracy.
> >
> > |  Eval Env  | History Information |  Step Accuracy↑  |
> > |:----------:|:-------------------:|:---------------:|
> > | val_unseen |   full              |     70.5%       |
> > | val_unseen |   3 steps           |     66.1%       |
> >
> >
> > ***Q4 Is there any bias in the synthetic data generation process?***
> >
> > Yes, the primary bias in the current pipeline is scene bias. Observing the rapid convergence when scaling synthetic data, as shown in Table 2, we infer that this occurs because we use only 10 real trajectories from a single scene to prompt the LLMs. This approach results in limited diversity and a scene bias in the generated instructions.
> >
> > For example, here are four generated instructions from the same output of the LLM:
> >
> > > *1. Start from the main entrance door, pass the living room, and enter the kitchen on your right. Locate the refrigerator, then turn left and stop just before the dining table.*
> > >
> > > *2. Navigate from the couch in the living room, move towards the mantel, and then stop next to the fireplace. Avoid any furniture and obstacles on your path.*
> > >
> > > *3. Begin at the foot of the bed in the master bedroom. Walk forward and enter the attached bathroom. Once you're inside, stop next to the bathtub.*
> > >
> > > *4. Start in the family room, walk towards the TV, then turn right and pass the bookshelf. Stop when you reach the large bay window overlooking the garden.*
> >
> > From the synthetic instructions provided above, we can observe that (1) the patterns of these instructions are similar, typically following a format like 'Start from place A, go past place B, stop at place C'; (2) the scenes are limited to living areas and a single floor. However, R2R tasks often require the agent to navigate across multiple floors and in various non-living areas.
> >
> > To address the scene bias during scaling up, one viable solution is to increase the diversity of the seed navigation instructions, which is relatively more cost-effective compared to obtaining a full trajectory. We conducted an experiment using 1,000 navigation instructions sampled from various R2R scenes to prompt GPT-4-turbo to generate 2,000 synthetic trajectories. These 2,000 trajectories are not constrained by the limitation that seed trajectories come from the same scene. The scaling results are presented in the table. Although the 2,000 trajectories generated by GPT-4-turbo are not of the same quality as those generated by GPT-4, scaling with them still outperforms the results from the 10,000-trajectory set.
> >
> > | Number of synthetic trajectories | Seed trajectories |         LLM         |  OSR↑ | SR↑   | SPL↑  |
> > |:--------------------------------:|:-----------------:|:-------------------:|:----:|------|------|
> > |               2,000              |         10        | GPT-4               | 42.2 | 31.1 | 26.6 |
> > |               2,000              |         1,000        | GPT-4-turbo               | 42.9 | 24.9 | 19.6 |
> > |           2,000 + 2,000          |     10 + 1,000    | GPT-4 + GPT-4-turbo | **43.2** | **32.6** | **28.3** |
> > |              10,000              |         10        | GPT-4               | 41.9 | 31.6 | 27.5 |
> >
> >
> > ***Q5 Minor clarifications/suggestions.***
> > We thank the reviewer for the suggestions. We will replace the term 'sim-to-real' with 'sim-to-sim.' The phrase 'real demonstration instruction and trajectory' refers to the instruction and the rollout of the episode, and we will clarify this in our revised version. We will also correct the typos in our subsequent version. Thank you again for pointing these out.
> >
> > [1] Hong, Yicong, et al. Vln bert: A recurrent vision-and-language bert for navigation.
> >
> > [2] Devlin J, Chang M W, Lee K, et al. Bert: Pre-training of deep bidirectional transformers for language understanding.

---

> ### Author Response · Authors · 2023-11-22
>
> Dear Reviewer Sthf,
>
> Thank you for your constructive comments and suggestions. As the discussion phase is nearing its end, we wondered if you might still have any concerns that we could address. We believe our response addressed all your questions/concerns, and hope that the work's impact and results are better highlighted with our responses. Thank you!
>
> Best wishes,
>
> Authors

---

> > ### Comment · Reviewer_Sthf · 2023-11-22
> > **Thanking authors for detailed response**
> >
> > I would like to thank the authors for their detailed response with clarifications and additional experiments. While both language and vision inputs are shown to have improved performance, the main hypothesis of the paper needs revision. The paper's claim has been further strengthened with more experiments (as suggested by reviewer PNyi), but the main challenges like (1) reliance on off-the-shelf object detectors/image captioning models vs training end-to-end policies (2) spatially grounding language descriptions vs hallucinations based on prior correlations, needs more investigation to use language as perceptual representation for realistic scenarios (not simulation). That's why I would maintain my borderline rating for now.

---

> ### Author Response · Authors · 2023-11-23
>
> We would like to thank the reviewer for expressing their concerns!
>
> For (1), we think the reliance on off-the-shelf object detectors/image captioning models is actually core feature of our approach. Since we disentangle the the image module and the language module, our approach can make use of independent advances in vision and language systems. As language models continue to improve (and as vision models continue to improve), LangNav can automatically (and immediately) make use of these advances. Also as noted by Reviewer 2DEF, language-guided visual navigation enjoys LLMs' generalization power and bridges the visual domain gap. Furthermore, it largely reduces the cost of generating synthetic navigation trajectories.
>
> For (2), we agree that the application of our LangNav system in realistic scenarios is essential! We actually conducted additional qualitative analysis on whether hallucinated text in non-standard environments can produce spatially grounded text. We indeed find that this is the case. See [here](https://docs.google.com/document/d/e/2PACX-1vRnsNCGDadAKsfYvpUD-q9dXGd8qGGkkJ04901S8ocE_52WE5ns_9C1KD_VX1LlznhcSmKqot3VNfxj/pub) for the examples, where we sampled a seed trajectory in a realistic office environment and then prompted GPT-4 to generate synthetic trajectories. We find that the synthetic trajectories are spatially consistent when attempting to draw the layout of the described environments (as also noted by the reviewer 2DEF). On the other hand, the R2R environment, despite being a simulation, is derived from actual household settings, complete with realistic image panoramas. [1] has shown that the policy learned in R2R could be adopted to the realistic environment.
>
> Do the above points change your view of the contribution at all?
>
> [1] Anderson et al. Sim-to-Real Transfer for Vision-and-Language Navigation.

---

### Official Review · Reviewer_PNyi · 2023-11-06

**Soundness:** 2 fair
**Presentation:** 3 good
**Contribution:** 1 poor
**Rating:** 3
**Confidence:** 4

**Summary:**

This paper tackles the task of Vision-and-Language Navigation (VLN) in the scarce data setting: defined to be small datasets with at most 100 instructions coupled with ground-truth trajectories. They propose to do this by decomposing VLN into visual perception and language-based decision-making, using language as the intermediate perceptual representation. Visual perception can be handled by existing vision-language models (VLMs), while a language-based decision-making policy can be fine-tuned from powerful language models (LMs). They design the representation as structured text prompts describing observations and trajectory history, which are extracted by VLMs. To train the LM policy, they synthesize a dataset by prompting Large Language Models (LLMs) to generate instructions and trajectories given the original small dataset given as an example. They conduct experiments to evaluate the performance of an LM policy fine-tuned on their synthesized data, and to evaluate how well an LM policy trained in simulation generalizes to real-world settings.

**Strengths:**

1. Experiments provide extensive comparisons across recent VLN baselines spanning a broad range of design choices: e.g. end-to-end learned systems, systems with explicit topological memory, systems using LLMs zero-shot for reasoning.

**Weaknesses:**

1. Evaluations do not directly address the main hypothesis. LangNav phrases its main focus as the hypothesis that a VLN system should use language as its perceptual representation (Sec. 1, para. 3). However, the paper’s evaluations focus on how language can help in low-data regimes rather than on the inherent merits of a language-based representation (e.g. an analysis of how well the representation captures and conveys information salient to decision-making, how easily and accurately the representation can be built with the proposed perception system etc.). I would suggest that the authors should clarify in the title and Secs. 1, 3 that their focus is on using language for VLN specifically in low-data situations.

2. Assumption of low-data setting for VLN in home environments seems contrived. Given rich and widely available datasets for VLN [1-4] and VLN-CE [5] tasks in household environments, the assumption on low data availability for VLN seems artificial. To better motivate this setting, the proposed system could also be tested on a wider variety of environments lacking extensive datasets (e.g. office environments, supermarkets, or industrial environments like workshops and warehouses). Since the inputs are purely language, it should not require excessive effort to hand-engineer a small number of trajectories based on real-world environments. These tests also serve to validate that LLMs can synthesise useful data in more exotic environments: while existing object-goal navigation works indicate LLMs certainly have learnt rich ‘common-sense’ about household environments, their capabilities in other environments remain less tested. I consider the need for strong motivation and evaluation of the low-data setting to be essential, especially given that the experiments indicate LangNav’s performance when trained on the entire dataset is still far from SOTA (Table 4).

3. Limited analysis provided on quality of proposed LLM-based data generation. Tables 1, 2 indicate that the LangNav agent trained on synthetic data converges to notably lower SR than an agent trained on the full R2R (Table 4), suggesting that LangNav’s generated synthetic data is poorer in some aspects compared to R2R. The authors should provide some insight into why this is the case, and more broadly explore the limitations of synthetic data generation using LLMs’ ‘common-sense’ as compared to data from hand-collected datasets like R2R, RxR etc.

4. Limited novelty or originality in using language for sim-to-real transfer. Similar observations and discussions on the effectiveness of using text or language intermediate representations to transfer a policy across environments have already been made in [6,7].

5. Limited exploration of visual perception and its coupling with the LM policy. LangNav largely assumes that existing VLMs enable VLN to be decomposed into visual perception and language-based decision-making, and thus consider only the decision-making aspect. However, failure modes such as the VLMs’ failure to identify features needed for LM policy to reason (Sec. 5) suggest that the coupling between the two components needs to be more deeply considered for robust VLN. E.g. Can VLMs be prompted also with navigation instructions or trajectory history to focus their attention on salient details, or can VLMs make use of some feedback from the LM policy?

[1] Yuankai Qi, Qi Wu, Peter Anderson, Xin Wang, William Yang Wang, Chunhua Shen, and Anton van den Hengel. Reverie: Remote embodied visual referring expression in real indoor environments. In CVPR, pages 9982–9991, 2020.

[2] Fengda Zhu, Xiwen Liang, Yi Zhu, Qizhi Yu, Xiaojun Chang, and Xiaodan Liang. Soon: Scenario oriented object navigation with graph-based exploration. In CVPR, pages 12689–12699, 2021.

[3] Peter Anderson, Qi Wu, Damien Teney, Jake Bruce, Mark Johnson, Niko Sunderhauf, Ian Reid, Stephen ¨ Gould, and Anton Van Den Hengel. Vision-and-language navigation: Interpreting visually-grounded navigation instructions in real environments. In CVPR, pages 3674–3683, 2018.

[4] Alexander Ku, Peter Anderson, Roma Patel, Eugene Ie, and Jason Baldridge. Room-across-room: Multilingual vision-and-language navigation with dense spatiotemporal grounding. In Proceedings of the 2020 Conference on Empirical Methods in Natural Language Processing (EMNLP), pages 4392–4412, 2020.

[5] Jacob Krantz, Erik Wijmans, Arjun Majumdar, Dhruv Batra, and Stefan Lee. Beyond the nav-graph: Vision-and-language navigation in continuous environments. In European Conference on Computer Vision, 2020.

[6] Mohit Shridhar, Xingdi Yuan, Marc-Alexandre Cote, Yonatan Bisk, Adam Trischler, and Matthew Hausknecht. 2021. ALFWorld: Aligning text and embodied environments for interactive learning. In International Conference on Learning Representations.

[7] Narasimhan, K., Barzilay, R., and Jaakkola, T. (2018). Grounding language for transfer in deep reinforcement learning. JAIR, 63(1):849–874.

**Questions:**

1. What are “gold” trajectories?

2. Given that there seems to be minimal constraints or priors placed on the trajectories/scenarios the LLM can generate (aside from several trajectory examples), how realistic are the outputs? Are the trajectories clearly in the desired household environments, and do they exhibit the expected spatial layout of the house?

3. Is the random action approach (Sec. 3.3) used for training the LangNav agent, and if it is, how is it used? This replanning approach is possible given a map of the environment. However, given that most of the data LangNav agents are trained on are trajectories in an environment hallucinated by an LLM, and which we have no map for, how is the random action approach used?

---

> ### Author Response · Authors · 2023-11-19
> **Response to Reviewer PNyi (1/N)**
>
> We want to thank Reviewer PNyi for these valuable comments.
>
> ***(1) Focus of our paper:***  We agree that more focus should be placed on the inherent merits of the language-based representation. We note that our two case studies (i.e., efficient synthetic data generation and ALFRED-to-R2R transfer) actually *are* explicitly leveraging the fact that language can serve as an efficient and robust representation of one's perception. However we agree that this aspect of our work should be emphasized, and will clarify this point more in the paper.
>
> Moreover, your suggestion of focusing on the inherit merits of language inspired us to perform a third "case study", where we experiment to see whether language-based representation allows us to *interpret* and *edit* our policy. Concretely, we randomly pick 10 R2R trajectories where the LangNav fails to choose the correct direction, like the second example in Figure 4 in the paper. For the step where the LangNav was incorrect, we inspected the captions, and when the captions were incorrect, we manually corrected them . We provide three examples which demonstrate that, by merely correcting the incorrect captions, LangNav is able to change the decision from incorrect to correct. The original trajectory examples are in [the link to original trajectories](https://docs.google.com/document/d/e/2PACX-1vRHHEyRJXVe0kHd6N4yMqe_XjQvHthB7VAs4f-NYNWBOH_-zDCfOtFtXAmhCs2ZPzHG-28QMHQKn_Vw/pub) and the modified trajectory examples are in [the link to modified trajectories](https://docs.google.com/document/d/e/2PACX-1vSUhL58LvLpF47YKDPcwcgk7rIHGhsW9IFOeW-pWkMlBg9v_8LaF4He-rmioksk-y-czpyxKIucKik6/pub), where we mark the modified caption in ***bold***. We applied this procedure to 10 selected R2R trajectories and discovered that our LangNav was able to revise its decision correctly in **70%** of these trajectories. For the other 30% examples, the failure reason doesn't fall into the vision captions in the current step. For example in Example #3, the agent decides to continue moving forward in Step 2 since it does not recognize from its historical data that it has already ascended the stairs.
>
> This case study highlights another benefit of working in language space---i.e., improved interpretability and editability. And we expect that LangNav will be able to leverage independent advances in vision-only and text-only models. We thank the reviewer for inspiring this experiment and will include it in the paper.
>
> ***(2) Testing in exotic environments:*** We thank the reviewer for pointing this out! We agree that the potential application in some exotic environments would be a strong motivation for LangNav. To validate that LLMs can synthesize useful data in more exotic environments, we conduct an experiment where we handcraft a trajectory in a real office environment and then prompt GPT-4 to generate synthetic trajectories within the scope of the office environment. Here are the sampled real trajectory and two generated synthetic trajectories: [link to real/synthetic trajectories in office environments](https://docs.google.com/document/d/e/2PACX-1vRnsNCGDadAKsfYvpUD-q9dXGd8qGGkkJ04901S8ocE_52WE5ns_9C1KD_VX1LlznhcSmKqot3VNfxj/pub).
>
> We can see that the synthetic trajectories (1) contain abundant common object-scene correlations in office environments, (2) exhibit great spatial consistency (example), and (3) incorporate a significant amount of captions and objects that do not directly relate to the given instruction. In summary, the LLM is capable of generating synthetic trajectories in the office environment with same quality to the household environment in Sec. 4.1.1. However, due to the lack of actual benchmarks for these types of exotic environments, we can't provide quantitative results beyond the household environment during this phase. We nonetheless think this is an interesting experiment and will update the paper with these qualitative results on exotic environments.

---

> ### Author Response · Authors · 2023-11-19
> **Response to Reviewer PNyi (2/N)**
>
> ***(3) Analysis on quality of synthetic data:*** Thanks for pointing this out. We infer that the fast convergence of synthetic data, compared to the **full realistic** dataset, is due to the use of only 10 real trajectories from a single scene to prompt LLMs.
>
>
> For example, here are four generated instructions from the LLM:
>
> > *1. Start from the main entrance door, pass the living room, and enter the kitchen on your right. Locate the refrigerator, then turn left and stop just before the dining table.*
> >
> > *2. Navigate from the couch in the living room, move towards the mantel, and then stop next to the fireplace. Avoid any furniture and obstacles on your path.*
> >
> > *3. Begin at the foot of the bed in the master bedroom. Walk forward and enter the attached bathroom. Once you're inside, stop next to the bathtub.*
> >
> > *4. Start in the family room, walk towards the TV, then turn right and pass the bookshelf. Stop when you reach the large bay window overlooking the garden.*
>
> We can see from the above synthetic instructions that (1) patterns of the synthetic instructions are similar, which are like "Start from place A, go pass place B, stop at place C", (2) scenes are limited to the living area and a single floor, however, the R2R tasks always require the agent navigating across floors and in some non-living area.
>
> In order to fairly compare the quality of the synthetic data and the real-world data, we conduct an experiment where we prepare 93 training trajectories for both real-world and synthetic data, and finetune LLaMA2-7B. The real-world trajectories all originate from the same scene used to sample the 10 trajectories for prompting LLMs. The evaluation results are shown in the table. The evaluation shows that, under the same constraints of the training scene diversity, synthetic trajectories have better quality than real-world trajectories. We infer the gain is from the more accurate and diverse language description of the scene.
>
> |  Eval Env  | Training data |   NE ↓ |   SR ↑  |  SPL ↑ |
> |:----------:|:-------------:|:-----:|:-----:|:-----:|
> |  val_seen  |   real-world  | 10.40 | 14.89 | 10.60 |
> |  val_seen  |    sythetic   | **8.81**  | **16.85** | **13.54** |
> | val_unseen |   real-world  | 10.16 | 14.09 |  9.54 |
> | val_unseen |    sythetic   | **8.29**  | **17.41** | **14.59** |
>
> This indicates that the underperformance of the generated trajectories compared to real trajectories is due to scene diversity. To further investigate the influence of the scene diversity, we conduct an experiment where we use 1000 navigation instructions sampled from various R2R scenes to prompt GPT-4-turbo to generate 2000 synthetic trajectories. The scaling results are listed in the table. We can see that although the 2000 trajectories generated by GPT-4-turbo are not of the same quality as those generated by GPT-4, scaling up using these trajectories outperforms the results from the 10000-trajectory set.
>
> | Number of synthetic trajectories | Seed trajectories |         LLM         |  OSR↑ | SR↑   | SPL↑  |
> |--------------------------------|-----------------|-------------------|----|------|------|
> |               2,000              |         10        | GPT-4               | 42.2 | 31.1 | 26.6 |
> |               2,000              |         1,000        | GPT-4-turbo               | 42.9 | 24.9 | 19.6 |
> |           2,000 + 2,000          |     10 + 1,000    | GPT-4 + GPT-4-turbo | **43.2** | **32.6** | **28.3** |
> |              10,000              |         10        | GPT-4               | 41.9 | 31.6 | 27.5 |
>
>
> ***(4) Novelty in sim2real transfer:*** We understand that LangNav is not the first work in learning high-level embodied knowledge through language. However, the differences between our sim2real experiments and [6,7] are fundamental.
>
> For [6], the agent learns the text policy from TextWorld [8] (which is extended to analog the ALFRED scenes) and then evaluates it in the ALFRED environment [9]. The learning and testing are in the two views (text & vision) of the same underlying world. While our sim2real experiments exhibit the ability of transferring knowledge between two very different underlying worlds: R2R and ALFRED. The significant differences are demonstrated in Figure 5.
>
> For [7], the agent learns to transfer the knowledge between two different game environments (Boulderchase and Bomberman). There are two main differences between their work and our sim2real transferring. First of all, the Boulderchase and the Bomberman are not designed for navigation tasks, and their environments are more simplistic and gamified compared to our testbeds (ALFRED and R2R). Thus, the difficulty level of transferring policies is different. Secondly, the transferring settings are different. In Section 3.4 of [7], it's noted that the transferred policy can further interact with and learn from the full new environment. While in our settings, we either do not allow or restrict the policy learned in ALFRED to only a few scenes in R2R.

---

> ### Author Response · Authors · 2023-11-19
> **Response to Reviewer PNyi (3/N)**
>
> ***(5) Exploration of visual perception:*** We strongly agree with the reviewer that more exploration in more accurate vision captions is needed to understand the system more deeply. As shown in third case study we conducted in (1), we show that we can manipulate the vision captions in the prompt and discover that inaccurate visual perceptions contribute significantly to the failure cases, which indicates that although the current VLM system is not yet perfect, we remain optimistic that future improvements in vision and text models will greatly enhance LangNav's long-term performance.
>
> To further explore how the language can help *on top of* continuous visual features, we performed further experiments where we extend the RecBert [9] by concatenating language features to the visual features to represent the candidate image. More specifically, the original RecBert uses ResNet-152 to extract the visual feature to represent the image, while our extension additionally concatenates the language feature of the caption to be the image representation. The captions are the same as what are used in LangNav and the features are extracted by BERT-base [10]. We pre-train and fine-tune the RecBert and its extension with 100-shot training trajectories and full R2R train set. The results are listed in the tables. We can see that the language features improve the performance in both few-shot and full training set cases, which indicates that language-based features can help the visual perception module for the VLN agent.
>
> | eval env   | train trajectories | perceptual features | SR↑  | SPL↑  |
> |------------|:--------------------:|:----------------------:|------|------|
> |  val_seen  |          100         |           vision   only         | 19.9 | 18.6 |
> | val_unseen |          100         |           vision  only         | 19.0 | 17.4 |
> |  val_seen  |          100         |           vision + language       |**22.8**|**21.5**|
> | val_unseen |          100         |           vision + language        |**19.3**|**18.0**|
>
> | eval env   | train trajectories  | perceptual features | SR↑   | SPL↑  |
> |------------|:--------------------:|:----------------------:|------|------|
> |  val_seen  |      full train      |            vision  only           | 58.5 | 55.4 |
> | val_unseen |      full train      |            vision  only           | 47.1 | 43.4 |
> |  val_seen  |      full train      |           vision + language        |**61.4**|**57.1**|
> | val_unseen |      full train      |           vision + language        |**48.8**|**44.1**|
>
> These results indicate that language as a perceptual representation can provide additional benefits on top of continuous visual features. We thank the reviewer for suggesting this experiment and will include it in the next version of the paper.
>
> ***Q1. What are "gold" trajectories?***
> The gold trajectories represent the optimal or expert-defined shortest path from the start point to the goal, as provided by the environment.
>
> ***Q2. How is the quality of the synthetic trajectory generated by LLMs?***
> The quality of the synthetic trajectory generated by LLMs is quite good. Quantitatively, we can see from the results in the response to (3) that within the same constraints, the synthetic trajectory is comparable to the real R2R trajectory. Qualitatively, as we demonstrated in the examples in Figure 3 and Section H, the synthetic trajectories are informative. The generated trajectories follow the desired instruction and exhibit consistent spatial layouts in each step. As we illustrated in Section 4.1.1, the synthetic trajectories have a strong prior knowledge base, spatial consistency, and are descriptive.
>
> ***Q3. How to apply the random approach to the synthetic trajectory?***
> Great question! The random approach is only applied to the R2R trajectories, and not the synthetic trajectories.
>
> [8] Côté M A, Kádár A, Yuan X, et al. Textworld: A learning environment for text-based games.
>
> [9] Hong, Yicong, et al. Vln bert: A recurrent vision-and-language bert for navigation.
>
> [10] Devlin J, Chang M W, Lee K, et al. Bert: Pre-training of deep bidirectional transformers for language understanding.

---

> > ### Comment · Reviewer_PNyi · 2023-11-23
> >
> > I would like to thank the authors for their responses addressing the weaknesses highlighted in the review, and would like to commend their effort in conducting various new experiments to bolster their results. In particular, their responses to weaknesses (2) and (3) help to further underscore the merits of their approach applied to a very low-data setting.
> >
> > However, the main concern highlighted in weakness (1) is still outstanding: that the paper pitches language as a general perceptual representation for VLN but does not adequately motivate this. To motivate this in updated versions of the paper, I believe the authors should explicitly assess how effective language is at representing salient information for VLN, how easily a language representation can be constructed, and evaluate its overall effect on VLN. There remains limited consideration of how an effective language representation can be constructed: the paper only considers using VLMs zero-shot for creating such representations, which can have limitations as highlighted in Sec. 5. Further, the authors’ response to weakness (5) seems to suggest value in using visual features in addition to language features, further diluting the argument that language is a general perceptual representation that can stand on its own.
> >
> > In summary, the paper still does not address its main hypothesis clearly. It focuses on more fringe benefits of using language as a representation, like synthesizing data with LLMs, transferability of language policies across environments, and editability of language representations. Some of these would be central benefits if the paper were to be reworked to focus on VLN in a very low-data regime. I would encourage the authors to recast their work in the low-data setting, as their experiments in that regard seem to be promising. I understand that the deadline for rebuttal is tight, and I might adjust my score during the reviewer discussion period. However, due to the paper's current lack of focus and clarity, I am unable to raise my score at this moment.

---

> ### Author Response · Authors · 2023-11-22
>
> Dear Reviewer PNyi,
>
> Thank you for your constructive comments and suggestions. As the discussion phase is nearing its end, we wondered if you might still have any concerns that we could address. We believe our response addressed all your questions/concerns, and hope that the work's impact and results are better highlighted with our responses. Thank you!
>
> Best wishes,
>
> Authors

---

> ### Author Response · Authors · 2023-11-23
>
> We would like to extend our sincere thanks to Reviewer PNyi for their timely reply before the deadline.
>
> In response to the reviewer's feedback, we acknowledge the need to enhance the clarity of our paper's focus and hypothesis. We aim to adjust our emphasis in the updated manuscript to highlight that:
>
> 1. Utilizing language as a perceptual representation is advantageous in the low-data setting due to its efficiency in synthesizing data with LLMs and its adaptability across various environments.
>
> 2. In contexts where training data is abundant, employing language as a perceptual representation not only maintains its efficacy but also amplifies visual perception capabilities, as demonstrated in our response to weakness (5).
>
> 3. The use of language as a perceptual representation offers superior interpretability, allowing for the editing and understanding of language to deduce reasons for failures.
>
> We plan to more clearly articulate these points in the revised version of our paper, hoping this refined emphasis will adequately address the concerns raised by the reviewer.

---

### Author Response · Authors · 2023-11-22
**Overall response**

We thank the reviewers for their helpful comments. We have performed extensive experiments based on the reviewers' suggestions and would like to highlight the main changes. (We also respond to the individual reviewer questions in the individual rebuttals).

**Practicality of LangNav**

Several reviewers have questioned the practicality of LangNav, given that (1) continuous visual features are likely to be almost always available, and that (2) language-based representation could be incomplete and ambiguous.

To address these concerns, we conducted an additional study where we use **both** vision and language as a perceptual representation by augmenting RecBERT to take text representations (given by a pretrained BERT-base model.) We find that under this setup, using language as a perceptual representation improves performance in **both low-data and full-data regimes**, as shown below.

| eval env   | train trajectories | perceptual features | SR↑  | SPL↑  |
|------------|:--------------------:|:----------------------:|------|------|
|  val_seen  |          100         |           vision   only         | 19.9 | 18.6 |
| val_unseen |          100         |           vision  only         | 19.0 | 17.4 |
|  val_seen  |          100         |           vision + language       |**22.8**|**21.5**|
| val_unseen |          100         |           vision + language        |**19.3**|**18.0**|

| eval env   | train trajectories  | perceptual features | SR↑   | SPL↑  |
|------------|:--------------------:|:----------------------:|------|------|
|  val_seen  |      full train      |            vision  only           | 58.5 | 55.4 |
| val_unseen |      full train      |            vision  only           | 47.1 | 43.4 |
|  val_seen  |      full train      |           vision + language        |**61.4**|**57.1**|
| val_unseen |      full train      |           vision + language        |**48.8**|**44.1**|

These results indicate that language as a perceptual representation can provide additional benefits on top of continuous visual features, even in non-low-data settings.

**Language representations for interpreting and editing the navigation policy**

Based on reviewer PNyi's suggestion, we investigated whether we could leverage the inherent interpretability and editability of discrete language. Concretely, we randomly pick 10 R2R trajectories where the LangNav fails to choose the correct direction, like the second example in Figure 4 in the paper. For the step where the LangNav was incorrect, we inspected the captions, and when the captions were incorrect, we manually corrected them . We provide three examples which demonstrate that, by merely correcting the incorrect captions, LangNav is able to change the decision from incorrect to correct. The original trajectory examples are in [the link to original trajectories](https://docs.google.com/document/d/e/2PACX-1vRHHEyRJXVe0kHd6N4yMqe_XjQvHthB7VAs4f-NYNWBOH_-zDCfOtFtXAmhCs2ZPzHG-28QMHQKn_Vw/pub) and the modified trajectory examples are in [the link to modified trajectories](https://docs.google.com/document/d/e/2PACX-1vSUhL58LvLpF47YKDPcwcgk7rIHGhsW9IFOeW-pWkMlBg9v_8LaF4He-rmioksk-y-czpyxKIucKik6/pub), where we mark the modified caption in ***bold***. We applied this procedure to 10 selected R2R trajectories and discovered that our LangNav was able to revise its decision correctly in **70%** of these trajectories. For the other 30% examples, the failure reason doesn't fall into the vision captions in the current step. For example in Example #3, the agent decides to continue moving forward in Step 2 since it does not recognize from its historical data that it has already ascended the stairs.

**Analysis of synthetic data quality**

We performed several investigations into our synthetic data quality, including a qualitative analysis on whether we can sample synthetic data from exotic environments, and how different mix of synthetic/real trajectories affect final performance.